# CHAMP: Conformalized 3D Human Multi-Hypothesis Pose Estimators

**Harry Zhang, Luca Carlone**
Massachusetts Institute of Technology
Cambridge, MA 02139, USA
`{harryz, lcarlone}@mit.edu`

## Abstract

We introduce CHAMP, a novel method for learning sequence-to-sequence, multi-hypothesis 3D human poses from 2D keypoints by leveraging a conditional distribution with a diffusion model. To predict a single output 3D pose sequence, we generate and aggregate multiple 3D pose hypotheses. For better aggregation results, we develop a method to score these hypotheses during training, effectively integrating conformal prediction into the learning process. This process results in a differentiable conformal predictor that is trained end-to-end with the 3D pose estimator. Post-training, the learned scoring model is used as the conformity score, and the 3D pose estimator is combined with a conformal predictor to select the most accurate hypotheses for downstream aggregation. Our results indicate that using a simple mean aggregation on the conformal prediction-filtered hypotheses set yields competitive results. When integrated with more sophisticated aggregation techniques, our method achieves state-of-the-art performance across various metrics and datasets while inheriting the probabilistic guarantees of conformal prediction. Interactive 3D visualization, code, and data will be available at this [website](website).

## 1 Introduction

Learning to estimate 3D human poses from videos is an important task in computer vision and robotics. Typical 3D human pose estimators learn to predict a single 3D human pose from an RGB image, either in an end-to-end manner or by relying on pre-existing 2D keypoint detection methods applied to the RGB images. Recent advances in sequence-to-sequence modeling enable learning 3D human poses in sequence by using RGB videos, which significantly improves the flexibility and efficiency of such estimators (Zhang et al., 2022b;a). While promising, such approaches only focus on *single-hypothesis* predictions, which is an inherently ill-posed problem when trying to reconstruct a 3D human pose given inputs collected from one viewpoint.

Thus, a more recent line of work focuses on learning *multi-hypothesis* 3D human pose estimators from 2D inputs. Instead of learning one deterministic target, such methods model the 2D-to-3D pose learning problem as learning a conditional distribution, which describes the distribution of the 3D poses given 2D inputs. This change in problem formulation prompts the use of generative models such as GANs, VAEs, and Diffusion models (Li & Lee, 2020; Sharma et al., 2019; Shan et al., 2023), which can propose multiple hypotheses of 3D poses given a single 2D input. For practical uses, one should also consider aggregating the hypotheses to generate one single output prediction. However, due to the imperfection of the trained generative models, using all hypotheses when aggregating could be inefficient and suboptimal if some are exceedingly inaccurate.

To counter the aforementioned problems, we take inspiration from an important tool in statistical learning, conformal prediction (CP) (Shafer & Vovk, 2008; Angelopoulos & Bates, 2021), which uses a *post-training* calibration step to guarantee a user-specified coverage: by allowing to predict confidence sets $C(X)$, CP guarantees the true value $Y$ to be included with confidence level $\alpha$, i.e. $P(Y \in C(X)) \geq 1 - \alpha$ when the calibration examples $(X_i, Y_i) \in I_{cal}$ are drawn exchangeably from the test distribution. There are typically two steps involved in the CP process: In the calibration step, the conformity scores on the calibration set are ranked to determine a cut-off threshold $\tau$ for the

predicted values guaranteeing coverage $1 - \alpha$, usually via quantile computation. In the prediction step, conformity scores, which measure the conformity between the output and possible ground-truth values, are computed to construct the confidence sets $C(X)$ by using the calibrated threshold $\tau$. CP

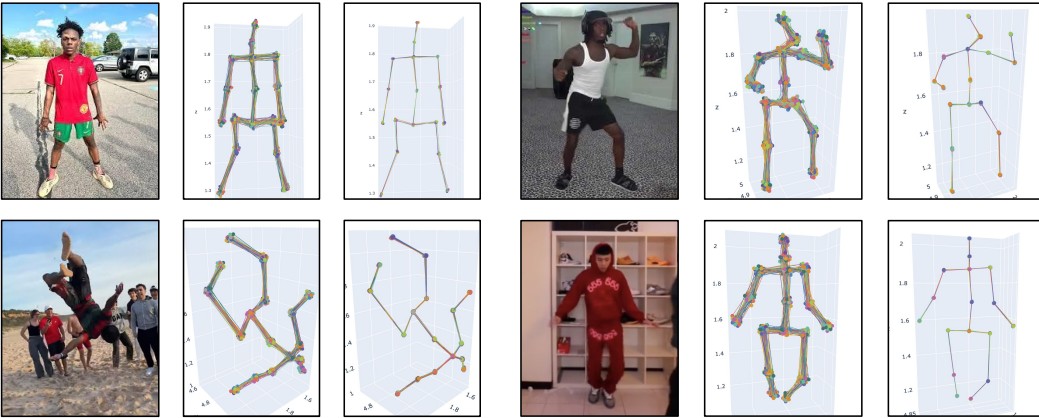

Figure 1: CHAMP sample results obtained on in-the-wild videos collected from TikTok. Having observed 2D keypoints, CHAMP proposes multiple hypotheses of the 3D human skeleton poses, and then a conformal predictor trained end-to-end with the pose estimator refines the confidence set by filtering out low-conformity-score hypotheses. This smaller set will be used in downstream aggregation for a single output prediction.

is highly flexible as it can be applied to any machine learning model. However, since it is applied post-training, learning the model parameters is not informed about the *post-hoc* conformalization step: the model is not tuned towards any specific CP objective such as reducing expected confidence set size (inefficiency). To bias the training towards lower inefficiency, ConfTr (Stutz et al., 2021) proposes a fully differentiable CP operation for classification tasks, which is applied end-to-end with the prediction model and optimizes a differentiable inefficiency loss in tandem with the original class loss for the classifier. This operation allows the model to learn parameters to reduce the inefficiency of the confidence set size during conformalization. Inspired by this, we wish to learn a scoring model that scores the "quality" of each hypothesis by measuring its conformity to the ground truth. We can then use this scoring model to simulate CP and optimize for the inefficiency in a differentiable manner, as done in (Stutz et al., 2021). During test time, we propose a large number of hypotheses and do regular CP to filter out the low-score ones before feeding into the downstream aggregation steps. The learned CP procedure can be applied to any multi-hypothesis estimator training process to refine the hypotheses confidence set size during test time for better aggregation results. With this learned CP wrapper, we present **CHAMP**, a **C**onformalized 3D **H**um**A**n **M**ulti-Hypotheses **P**ose Estimator. To summarize, the contributions of this work include:

- A novel sequence-to-sequence, multi-hypothesis 3D human pose estimator from 2D keypoints.
- A novel method to conformalize 3D human pose estimates during training by learning a score function to rank the quality of the proposed hypotheses.
- A novel method to aggregate the multiple hypotheses from the estimator output using conformalization based on a learned score function.
- Quantitative and qualitative results that demonstrate the state-of-the-art results of our method on a variety of real-world datasets.

## 2   RELATED WORK

**Diffusion Models** are a family of generative models that gradually corrupt data by adding noise, and then learn to recover the original data by reversing this process (Sohl-Dickstein et al., 2015; Ho et al., 2020; Song & Ermon, 2020; Song et al., 2020b;a). They have achieved success in generating high-fidelity samples in various applications such as image generation (Ho et al., 2022; Saharia et al., 2022a; Nichol et al., 2021; Batzolis et al., 2021; Rombach et al., 2022; Saharia et al., 2022b), multi-modal learning (Levkovitch et al., 2022; Kim et al., 2022; Huang et al., 2022; Avrahami et al., 2022; Fan et al., 2023) and pose estimations in 3D vision problems (Gong et al., 2023; Wu et al., 2023). We use Diffusion models to learn 3D human poses from 2D keypoints.

**3D Human Pose Estimation** is an important problem in computer vision and robotics. With deep learning, various end-to-end approaches have been proposed (Tekin et al., 2016; Pavlakos et al., 2017; Sun et al., 2018). However, with the maturity of 2D human keypoints detection, (Ho et al., 2022), more robust approaches focus on uplifting 2D keypoints to 3D, resulting in better performance (Xu & Takano, 2021; Zhao et al., 2019; Ma et al., 2021), where one typically deals with a frame-to-frame problem, or a sequence-to-sequence problem. Being able to predict a 3D keypoints sequence directly from a 2D keypoint sequence is highly desirable as it has higher efficiency and flexibility. In this scheme, deterministic methods learn to predict one single 3D output from the 2D input (Zhan et al., 2022; Zhang et al., 2022b; Zhao et al., 2023). However, with a single viewpoint, such a formulation is ill-posed because there could be many possible 3D poses satisfying the same 2D keypoint configuration. Thus, it is desirable to model the problem as a conditional probability distribution. Deep generative models have shown great potential in modeling such distributions. Specifically, mixed-density network (Li & Lee, 2019), VAE (Sharma et al., 2019), normalizing flows (Wehrbein et al., 2021), GAN (Li & Lee, 2019), and Diffusion models (Holmquist & Wandt, 2023; Shan et al., 2023) have all been applied to modeling such conditional distribution. In our work, we use (Zhang et al., 2022b) as the backbone for generating 3D keypoints sequences from 2D keypoints sequences. To infer multiple hypotheses, we follow the frameworks in (Shan et al., 2023; Gong et al., 2023; Zhou et al., 2023) to learn to recover 3D pose hypotheses from Gaussian noises. Ci et al. (2023) uses an innovative approach to learning probabilistic human poses via gradient flow diffusion. Zhu et al. (2023) proposes a novel framework for scaling up 2D-3D human pose learning. Mehraban et al. (2024) models local dependencies inherent in human pose sequences, outperforming transformers backbones. Xu et al. (2024) proposes multiple hypotheses with Diffusion and learns to score them end-to-end, but it does not consider the inefficiency of the prediction set during training.

**Conformal Prediction** (CP) is a powerful and flexible distribution-free uncertainty quantification technique that can be applied to any machine learning model (Angelopoulos & Bates, 2021; Shafer & Vovk, 2008). Assuming the exchangeability of the calibration data, CP has desirable coverage guarantees. Thus, it has been applied to many fields such as robotics (Huang et al., 2023; Lindemann et al., 2023), pose estimation (Yang & Pavone, 2023), and image regression (Teneggi et al., 2023; Angelopoulos et al., 2022). More sophisticated CP paradigms have been also proposed to tackle distribution shift and online learning problems (Gibbs & Candès, 2022; Bhatnagar et al., 2023). Since CP is applied post-training, the learned model is not aware of such conformalization during training. To gain better control over the confidence set of CP, learning conformal predictors and nonconformity score end-to-end with the model has been proposed in (Stutz et al., 2021; Fisch et al., 2021; Bai et al., 2022). More closely related to our proposal is the work of (Stutz et al., 2021), but instead of using raw logits as the conformity score for classification, we learn an extra scoring model as the score to rank the hypotheses and simulate CP during training.

## 3 PROBLEM FORMULATION

We are interested in the problem of learning a sequence of 3D human poses from a sequence of 2D human pose keypoints. We assume the 2D keypoints are available to us, which could be detected from the RGB images using well-established methods such as (Li et al., 2022b). Formally, given the input 2D keypoints sequence $\boldsymbol{x} = \{\boldsymbol{p}_n^{2d} | n = 1, \ldots, N\}$, where $\boldsymbol{p}_n^{2d} \in \mathbb{R}^{J \times 2}$, our goal is to learn a conditional distribution for the possible corresponding 3D positions of all joints $p_\theta(\boldsymbol{y}|\boldsymbol{x})$, where the sequence $\boldsymbol{y} = \{\boldsymbol{p}_n^{3d} | n = 1, \ldots, N\}$ and $\boldsymbol{p}_n^{3d} \in \mathbb{R}^{J \times 3}$. Here, $N$ represents the number of input and output frames and $J$ is the number of human joints in each frame. With the learned distribution $p_\theta(\boldsymbol{y}|\boldsymbol{x})$, we are able to do inference and generate hypotheses of 3D poses $\boldsymbol{H_y} \in \mathbb{R}^{H \times N \times J \times 3}$, where $H$ is the number of the hypotheses. We then conformalize the hypotheses to select the higher-quality ones and aggregate the latter in order to obtain a single estimate $\tilde{\boldsymbol{y}}$ as the final output.

## 4 METHODS

We use a *Diffusion Model* (Ho et al., 2020) to learn the conditional distribution $p_\theta(\boldsymbol{y}|\boldsymbol{x})$ due to its capability of modeling complex distributions and producing high-quality samples. With the trained Diffusion Model, we are able to generate many hypotheses sequences and aggregate them to produce a robust single prediction sequence for the final output. To achieve this, we learn an extra scoring model to rank the hypotheses and optimize the size of the hypotheses confidence set, thresholded by

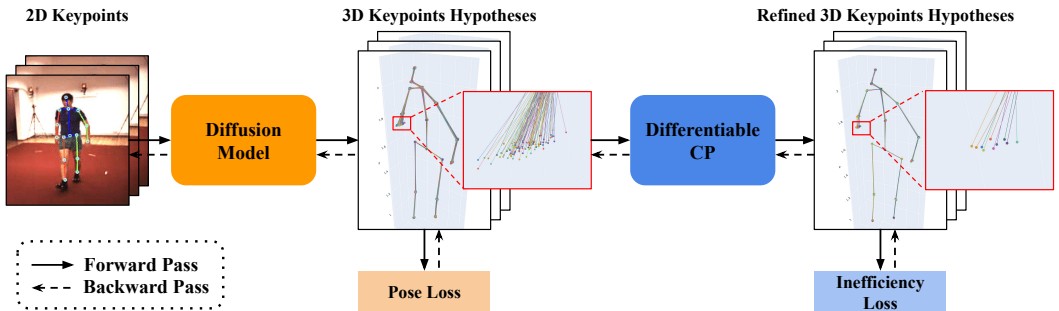

Figure 2: CHAMP Overview. CHAMP takes as input a sequence of 2D keypoints detected on a series of input RGB video frames. The 2D keypoints sequence gets fed into a Diffusion Model to produce 3D keypoints hypotheses for the sequence. The output of the Diffusion Model is supervised via a Pose Loss. Then we apply differentiable CP end-to-end during training on the hypotheses sequences, resulting in a smaller confidence set. The confidence set is used to calculate an Inefficiency Loss during training. Note that we show one frame in the sequence and hard assignment for the confidence set during training for better interpretability.

the 90% quantile score, effectively simulating conformal prediction during training, as illustrated in Fig. 2.

## 4.1 LEARNING 3D HUMAN POSES WITH A DIFFUSION MODEL

The use of Diffusion Models in 3D human pose estimation has been shown effective in various previous works (Feng et al., 2023; Gong et al., 2023; Shan et al., 2023; Choi et al., 2023), and we adapt the same paradigm in modeling the forward and the reverse process for the Diffusion Model.

**Forward Process.** In the forward process, the Diffusion Model takes as input a ground-truth 3D human pose sequence $\boldsymbol{y}_0 \in \mathbb{R}^{N \times J \times 3}$ and a sampled time step $t \sim \text{Unif}(0, T)$, where $T$ is the maximum number of diffusion steps. Then the forward process gradually diffuses the input by adding independent Gaussian noises $\boldsymbol{\epsilon} \sim \mathcal{N}(0, \boldsymbol{I})$ at each step. The nice property reported in (Ho et al., 2020) shows that the process can be succinctly written as:

$$q(\boldsymbol{y}_t|\boldsymbol{y}_0) = \sqrt{\bar{\alpha}_t}\boldsymbol{y}_0 + \sqrt{1 - \bar{\alpha}_t}\boldsymbol{\epsilon} \qquad (1)$$

where $\bar{\alpha}_t$ is a constant depending on the cosine variance schedule.

**Reverse Process.** To train such a Diffusion Model, in the reverse process, we denoise the corrupted 3D pose sequence $\boldsymbol{y}_t$. While the derivation in (Ho et al., 2020) simplifies the ELBO objective to minimizing the distance between the injected noise $\boldsymbol{\epsilon}_t$ and a learned noise $\boldsymbol{\epsilon}_\theta(\sqrt{\bar{\alpha}_t}\boldsymbol{y}_0 + \sqrt{1 - \bar{\alpha}_t}\boldsymbol{\epsilon}_t, t)$, we frame the problem slightly differently, following (Shan et al., 2023). Instead of training a network to learn the injected noise, we train a denoiser model $D_\theta$ that outputs the predicted pose $\bar{\boldsymbol{y}}_0$ directly:

$$\bar{\boldsymbol{y}}_0 = D_\theta(\boldsymbol{y}_t, \boldsymbol{x}, t) \qquad (2)$$

where the denoiser model takes as input the corrupted 3D pose sequence, the input 2D keypoints sequence, and the diffusion step to recover the uncorrupted input pose sequence $\bar{\boldsymbol{y}}_0$. We then supervise the predicted sequence with the ground-truth sequence using the mean per-joint MSE:

$$\mathcal{L}_{\text{pose}} = \frac{||\boldsymbol{y}_0 - \bar{\boldsymbol{y}}_0||_2^2}{N \cdot J} \qquad (3)$$

**Denoiser Model.** We build upon the MixSTE model (Zhang et al., 2022b) as the denoiser model in the reverse process of the Diffusion Model. MixSTE uses two separate attention mechanisms, effectively learning spatial and temporal relationships of the keypoints sequence in a modular way by combining several spatial and temporal attention blocks. To condition the MixSTE network on the corrupted 3D keypoints sequence, we change the network input by concatenating 2D keypoints and noisy 3D poses and we add a diffusion timestep embedding after the input embedding in the first attention block. A detailed description of the denoiser model is given in Appendix B and D.

**Generating Hypotheses.** Using the denoiser model, which models the conditional distribution $p_\theta(\boldsymbol{y}|\boldsymbol{x})$, we are able to generate a number of hypotheses during inference. Following the reverse

process of the Diffusion Model, we obtain the hypotheses set $\boldsymbol{H_y} = \{\bar{\boldsymbol{y}}^1, \bar{\boldsymbol{y}}^2, \ldots, \bar{\boldsymbol{y}}^H\}$ of size $H$ by sampling standard normal noises $\boldsymbol{y}_T^h \sim \mathcal{N}(0, \boldsymbol{I}) \in \mathbf{R}^{N \times J \times 3}$, where the superscript $h = [1, 2, \ldots, H]$ indexes the hypotheses, and feeding them into the denoiser model $D_\theta$:

$$\bar{\boldsymbol{y}}^h = D_\theta(\boldsymbol{y}_T^h, \boldsymbol{x}, T) \tag{4}$$

## 4.2 Learning Conformalization for the Hypotheses Confidence Set

While we are able to generate arbitrarily large numbers of hypotheses during test time with the distribution learned by the Diffusion Model, we still need to aggregate the proposed hypotheses into a single prediction. The correct way to aggregate the hypotheses remains an open problem. Taking a naive average would be suboptimal as the existence of outliers in the proposed hypotheses might skew the average. Other approaches improve the aggregation step by selecting the hypotheses that result in closer distances to 2D keypoints after projection via camera matrices, but still retain a large set of hypotheses before the aggregation step (Shan et al., 2023).

**Conformal Prediction.** A possible solution is to *conformalize* the hypotheses using conformal prediction (CP). Conformal prediction produces a set of predictions which covers the ground truth with high probability: given a trained prediction model $f_\theta$, conformal prediction first calibrates a cutoff value $\tau$ by ranking a calibration set $I_{\text{cal}}$ given each component's *conformity score* $\phi(\boldsymbol{y}_{\text{cal}})$ and setting the cutoff value as the user-specified $(1 + 1/|I_{\text{cal}}|)\alpha$-quantile of the calibration set scores. After the calibration step (conditioned on the trained model), the conformal predictor constructs the conformal prediction set as follows:

$$C_\theta(\boldsymbol{x}, \tau) = \{\boldsymbol{y} : \phi_\theta(\boldsymbol{y}) \geq \tau\} \tag{5}$$

With a properly selected conformity score function $\phi$, as well as a user-specified coverage parameter $\alpha$, the **post-training** conformal prediction guarantees the true value's conformity score has a set membership in $C_\theta(\boldsymbol{x}, \tau)$ with confidence level $\alpha$, i.e., $P(\boldsymbol{y}_{\text{GT}} \in C_\theta(\boldsymbol{x}, \tau)) \geq \alpha$. A key metric for a conformal predictor is the size of the confidence set $C_\theta(\boldsymbol{x}, \tau)$, or *inefficiency*: a good confidence set should be large enough to cover the ground truth, and small enough to be informative (low inefficiency). A very large confidence set could cover the ground truth with high probability, but it might not be necessarily useful as it is inefficient to consider a lot of possible values (high inefficiency). Given $H$ hypotheses of 3D human pose sequences, we wish to use the threshold conformal predictor above to refine the hypotheses set, resulting in $H'$ "good" hypotheses, where $H' \leq H$. Essentially, we wish to have a *low-inefficiency* conformal predictor, which selects a smaller number of elite hypotheses from the set for downstream aggregation. CP is intended to be used as a "wrapper" around the prediction model, and we wish to achieve better, fine-grained control over the inefficiency of the CP wrapper for downstream aggregation, conditioned on the input 2D and output 3D human pose sequence data. Thus, inspired by (Stutz et al., 2021), we integrate CP into the training, optimizing for the inefficiency of the confidence set online by simulating CP **during training** in a differentiable manner.

**Learning CP via Inefficiency Optimization.** We would like to make the model itself aware of the *post-hoc* hypotheses aggregation step so that the aggregation step does not get skewed by low-quality hypotheses. Thus, there should be an extra component in the model to shape the hypotheses confidence set. We build on top of the ConfTr paradigm proposed in (Stutz et al., 2021). We first design a conformity score function for poses. Unlike (Yang & Pavone, 2023), where the score function $\phi$ is hand-designed, we wish to learn a conformity score function $\phi_\theta$ that measures the distance between the proposed hypothesis and the ground truth. During training, we propose $H_{\text{train}}$ predicted output sequences as the hypotheses for each input 2D keypoints sequence $\boldsymbol{x}$ in the mini-batch. To implement the score function, we reuse the input embedding layer of the modified MixSTE and further project the embeddings (conditioned on input 2D keypoints $\boldsymbol{x}$) into two scalar values with an extra MLP that predicts a value $\in [0, 1]$ representing the probability that the hypothesis embedding belongs to the manifold of plausible 3D human pose embeddings. The conformity score function is implemented as a discriminator-style scoring function that measures the quality of the generated 3D keypoints (Kocabas et al., 2020). Thus, a higher score function output means the hypothesis is more likely (more realistic) to be from ground-truth 3D pose sequence distribution in the embedding space. Formally, the scoring function optimizes the following loss:

$$\mathcal{L}_S = \mathbb{E}\left[(S_\theta(\Theta_{\text{GT}}) - 1)^2\right] + \mathbb{E}\left[S_\theta(\bar{\Theta}^h)^2\right], \tag{6}$$

where $\Theta_{\text{GT}}$ and $\bar{\Theta}^h$ are the input embeddings of ground-truth 3D poses and generated 3D poses conditioned on the 2D keypoints $\boldsymbol{x}$, and $S_\theta$ is the scoring function MLP. We also add an adversarial loss that will be back-propagated into the denoiser model, as done in Kocabas et al. (2020); Goodfellow et al. (2014), which encourages the 3D pose prediction model to output more realistic samples:

$$\mathcal{L}_{\text{adv}} = \mathbb{E}\left[(S_\theta(\bar{\Theta}^h) - 1)^2\right] \tag{7}$$

Lastly, during training, we use the output of $S_\theta$ as the the conformity score between the hypothesis 3D pose sequence $\bar{\boldsymbol{y}}^h$ and the ground-truth 3D pose sequence $\boldsymbol{y}_{\text{GT}}$:

$$\phi_\theta(\bar{\boldsymbol{y}}^h) = S_\theta(\bar{\Theta}^h). \tag{8}$$

With the conformity score model output, we perform differentiable CP on each mini-batch of size $B$ during training. Similar to (Stutz et al., 2021), we split each set of $H_{\text{train}}$ hypotheses of the mini-batch in half: the first half, $\boldsymbol{H}_{\text{cal}}$, is used for calibration, and the second half, $\boldsymbol{H}_{\text{pred}}$, for prediction and loss computation. On $\boldsymbol{H}_{\text{cal}}$, we calibrate $\tau$ by computing the $(1 + 1/|\boldsymbol{H}_{\text{cal}}|)\alpha$-quantile of the conformity scores in a differentiable manner. On $\boldsymbol{H}_{\text{pred}}$, we calculate the soft inefficiency score by using the *soft* assignment of hypothesis $\bar{\boldsymbol{y}}^h$ being in the prediction set given the quantile threshold $\tau$:

$$\tilde{C}_\theta(\boldsymbol{x}, \tau) := \sigma\left(\frac{\phi_\theta(\bar{\boldsymbol{y}}^h) - \tau}{\eta}\right) \tag{9}$$

where $\sigma$ here is the sigmoid function and $\eta$ is the temperature hyperparameter. When $\eta \to 0$, we recover the *hard* assignment and $\tilde{C}_\theta(\boldsymbol{x}, \tau) = 1$ if $\phi_\theta(\bar{\boldsymbol{y}}^h) \geq \tau$ and 0 otherwise[1]. Thus, we are able to measure the size of the CP set of the hypotheses by $\sum_h \tilde{C}_\theta(\boldsymbol{x}, \tau)$. On $\boldsymbol{H}_{\text{pred}}$, we compute $\tilde{C}_\theta(\boldsymbol{x}, \tau)$ and then calculate the differentiable inefficiency score:

$$\Omega\left(\tilde{C}_\theta(\boldsymbol{x}, \tau)\right) = \max\left(\sum_{h=1}^{|\boldsymbol{H}_{\text{pred}}|} \tilde{C}_\theta(\boldsymbol{x}, \tau) - \kappa, 0\right) \tag{10}$$

where $\kappa$ is a hyperparameter that avoids penalizing singletons. With the differentiable inefficiency score defined, we are able to optimize the expected prediction set size across batches during training:

$$\mathcal{L}_{\text{size}} = \log \mathbb{E}[\Omega(\tilde{C}_\theta(\boldsymbol{x}, \tau))] \tag{11}$$

Combined with the adversarial loss, we have the "inefficiency" loss:

$$\mathcal{L}_{\text{ineff}} = \mathcal{L}_{\text{size}} + \mathcal{L}_{\text{adv}}. \tag{12}$$

**Full Training Objective.** The computational graph of this loss involves the denoiser model input embedding, and since every operation is made differentiable by construction, we are able to back-propagate the inefficiency loss into the model, further shaping the model on top of the original pose loss. The final training objective of our model is thus:

$$\mathcal{L} = \mathcal{L}_{\text{pose}} + \lambda\mathcal{L}_{\text{ineff}} \tag{13}$$

where $\lambda$ is the weight for combined the inefficiency loss.

### 4.3 CONFORMALIZED INFERENCE

During inference, any Conformal Prediction method can be applied to re-calibrate $\tau$ on a held-out calibration set $I_{\text{cal}}$ as usual, i.e., the thresholds $\tau$ obtained during training is not kept during test time. Given a trained pose estimation denoiser model $D_\theta$ and the conformity scoring model $\phi_\theta$ trained end-to-end with $D_\theta$, for each testing 2D keypoints sequence, $\boldsymbol{x}$, we wish to construct a hypotheses confidence set for the downstream aggregation as follows:

$$C_\theta(\boldsymbol{x}, \tau) = \{\bar{\boldsymbol{y}} : \phi_\theta(\bar{\boldsymbol{y}}) \geq \tau\}. \tag{14}$$

To generate candidate hypotheses for the hypotheses confidence set, we sample from the learned distribution $p_\theta(\boldsymbol{y}|\boldsymbol{x})$ $H$ times using (4) and refine them using (14), resulting in a smaller hypotheses set of size $H'$. We then aggregate the set $C_\theta(\boldsymbol{x}, \tau)$ by using a weighted average based on the conformity score output values to obtain a single output for practical use:

$$\bar{\boldsymbol{y}}_{\text{output}} = \frac{\sum_{\bar{\boldsymbol{y}} \in C_\theta(\boldsymbol{x}, \tau)} \phi_\theta(\bar{\boldsymbol{y}}) \cdot \bar{\boldsymbol{y}}}{\sum_{\bar{\boldsymbol{y}} \in C_\theta(\boldsymbol{x}, \tau)} \phi_\theta(\bar{\boldsymbol{y}})} \tag{15}$$

---

[1]Note that we disambiguate prediction sets from hard and soft assignments respectively using $C_\theta$ and $\tilde{C}_\theta$.

## 5 EXPERIMENTS

To evaluate our method, we train and test on standard human pose estimation datasets and provide quantitative results. We also provide qualitative results on in-the-wild videos.

**Human3.6M** (Ionescu et al., 2013) is the standard indoor dataset for 3D human pose estimation. The dataset collects videos from 11 actors engaging in 15 activities and the pose sequence videos are captured by 4 synchronized and calibrated cameras at 50Hz. Similar to (Shan et al., 2023; Zhang et al., 2022b), we train on 5 actors (S1, S5, S6, S7, S8) and evaluate on 2 actors (S9, S11). Quantitatively, following the standard evaluation scheme, we report the mean per joint position error (MPJPE), which is often referred to as Protocol #1, which computes the mean Euclidean distance between estimated and ground truth 3D joint positions in millimeters. We also provide Protocol #2 results in Appendix E.

**MPI-INF-3DHP** (Mehta et al., 2017) is a more challenging dataset consisting of indoor and outdoor activities, from 14 camera views which cover a greater diversity of poses. The training set contains 8 activities, and the test set contains 6. We preprocess the dataset with the valid frames provided by the authors for testing, following (Zhang et al., 2022b; Shan et al., 2023; Gong et al., 2023). Quantitatively, we report MPJPE, the percentage of correct keypoints (PCK), which describes the percentage of keypoints with Euclidean error less than 150mm, as well as the AUC score for this percentage.

For both of the datasets, in our setting, we hold out 2% of the training dataset for conformal calibration during test time. The held-out calibration set is not seen by the network during training. We compare 4 variations of our method. The backbone (weights) of the variations and the training process remain the same, but the key difference lies in the aggregation step during test time:

- **CHAMP-Naive**: Trained with inefficiency loss, but for multi-hypothesis scenarios, during test time, we do not refine the hypotheses set with CP, and we use all hypotheses for aggregation. In single-hypothesis scenarios, we only propose 1 hypothesis as the output.
- **CHAMP-Naive-Agg**: CHAMP-Naive but aggregated with the J-Agg method from (Shan et al., 2023) instead of taking a simple average.
- **CHAMP**: Same backbone as CHAMP-Naive, but only aggregates refined hypotheses confidence set in eq. (14) via weighted mean aggregation.
- **CHAMP-Agg**: Instead of taking the average, we use the J-Agg method from (Shan et al., 2023) on the set in eq. (14), which uses known or estimated intrinsic camera parameters to reproject 3D hypotheses to the 2D camera plane and selecting the joint hypotheses with the minimum reprojection error.
- **CHAMP-Best**: We use J-Best method in (Shan et al., 2023) on the set in eq. (14), which selects the joint hypothesis that is closest to the ground truth, and then combines the selected joints into the final 3D pose. This is the upper bound of J-Agg performance.

### 5.1 RESULTS ON HUMAN-3.6M

We discuss the quantitative results on the Human-3.6M dataset. In the single-hypothesis setting, we set $H = 1$ to compare with other deterministic methods. While our method primarily focuses on multi-hypothesis scenarios, we propose 1 hypothesis and use the CHAMP-Naive variation, without using the conformal prediction pipeline. As the results suggest in Table 1, our method achieves performance on par with the current state-of-the-art methods in the single-hypothesis case.

In the multi-hypothesis setting, we set $H = 80$. CHAMP variants achieve SOTA results, especially when combined with more sophisticated aggregations proposed in D3DP (Shan et al., 2023). On average, without the conformal prediction pipeline, CHAMP-Naive is able to achieve 39.2mm MPJPE error and improves by 2.3mm when using conformal prediction and mean aggregation. Using J-Agg and J-Best, CHAMP gets further improved by 2.1mm and 4.0mm respectively.

### 5.2 RESULTS ON MPI-INF-3DHP

We discuss the results on the MPI-INF-3DHP dataset in Table 2. In the single-hypothesis setting, CHAMP-Naive achieves results on par with other deterministic methods for the three metrics.

| Mean Per-Joint Position Error (Protocol # 1) - mm | | | | | | | | | | | | | | | |
|---|---|---|---|---|---|---|---|---|---|---|---|---|---|---|---|
| | AVG. | Dir | Disc. | Eat | Greet | Phone | Photo | Pose | Pur. | Sit | SitD. | Smoke | Wait | WalkD. | Walk | WalkT. |
| **Single Hypothesis** | | | | | | | | | | | | | | | | |
| Ray3D (Zhan et al., 2022) | 49.7 | 44.7 | 48.7 | 48.7 | 48.4 | 51.0 | 59.9 | 46.8 | 46.9 | 58.7 | 61.7 | 50.2 | 46.4 | 51.5 | 38.6 | 41.8 |
| STE (Li et al., 2022a) | 43.6 | 39.9 | 43.4 | 40.0 | 40.9 | 46.4 | 50.6 | 42.1 | 39.8 | 55.8 | 61.6 | 44.9 | 43.3 | 44.9 | 29.9 | 30.3 |
| P-STMO (Shan et al., 2022) | 42.8 | 38.9 | 42.7 | 40.4 | 41.1 | 45.6 | 49.7 | 40.9 | 39.9 | 55.5 | 59.4 | 44.9 | 42.2 | 42.7 | 29.4 | 29.4 |
| MixSTE (Zhang et al., 2022b) | 41.0 | 37.9 | 40.1 | 37.5 | 39.4 | 43.3 | 50.0 | 39.8 | 39.9 | 52.5 | 56.6 | 42.4 | 40.1 | 40.5 | 27.6 | 27.7 |
| DUE (Zhang et al., 2022a) | 40.6 | 37.9 | 41.9 | 36.8 | 39.5 | 45.6 | 49.2 | 40.8 | 40.1 | 40.7 | 47.9 | 53.5 | 42.4 | 41.1 | 40.3 | 30.8 | 28.6 |
| D3DP (Shan et al., 2023) | 40.0 | 37.7 | 39.9 | 35.7 | 38.2 | 41.9 | 48.8 | 39.5 | 38.3 | 50.5 | 53.9 | 41.6 | 39.4 | 39.8 | 27.4 | 27.5 |
| **CHAMP-Naive** (H=1) | 39.7 | 37.8 | 40.0 | 35.7 | 38.1 | 41.5 | 48.7 | 39.2 | 38.2 | 50.1 | 53.3 | 41.5 | 38.9 | 39.7 | 26.7 | 27.2 |
| **Multiple Hypotheses** | | | | | | | | | | | | | | | | |
| MHFormer (Li et al., 2022b) | 43.0 | 39.2 | 43.1 | 40.1 | 40.9 | 44.9 | 51.2 | 40.6 | 41.3 | 53.5 | 60.3 | 43.7 | 41.1 | 43.8 | 29.8 | 30.6 |
| GraphMDN (Oikarinen et al., 2021) | 61.3 | 51.9 | 56.1 | 55.3 | 58.0 | 63.5 | 75.1 | 53.3 | 56.5 | 69.4 | 92.7 | 60.1 | 58.0 | 65.5 | 49.8 | 53.6 |
| DiffuPose (Choi et al., 2023) | 49.4 | 43.4 | 50.7 | 45.5 | 50.2 | 49.6 | 53.4 | 48.6 | 45.0 | 56.9 | 70.7 | 47.8 | 48.2 | 51.3 | 43.1 | 43.4 |
| DiffPose (Gong et al., 2023) | 36.9 | 33.2 | 36.6 | 33.0 | 35.6 | 37.6 | 45.1 | 35.7 | 35.5 | 46.4 | 49.9 | 37.3 | 35.6 | 36.5 | 24.4 | 24.1 |
| DiffPose (Feng et al., 2023) | 43.3 | 38.1 | 43.3 | 35.3 | 43.1 | 46.6 | 48.2 | 39.0 | 37.6 | 51.9 | 59.3 | 41.7 | 47.6 | 45.5 | 37.4 | 36.0 |
| D3DP (Shan et al., 2023) | 39.5 | 37.3 | 39.5 | 35.6 | 37.8 | 41.4 | 48.2 | 39.1 | 37.6 | 49.9 | 52.8 | 41.2 | 39.2 | 39.4 | 27.2 | 27.1 |
| **CHAMP-Naive** | 39.2 | 37.1 | 39.1 | 35.1 | 37.1 | 41.4 | 48.1 | 39.0 | 36.9 | 49.4 | 52.0 | 40.6 | 38.9 | 38.9 | 27.3 | 27.1 |
| **CHAMP-Naive-Agg** | 38.7 | 36.8 | 38.9 | 34.4 | 36.4 | 41.0 | 47.2 | 38.7 | 36.5 | 48.8 | 51.2 | 40.2 | 38.4 | 38.2 | 27.3 | 26.5 |
| **CHAMP** | 36.9 | 36.2 | 38.4 | 32.3 | 33.2 | 39.1 | 43.9 | 36.2 | 34.5 | 44.9 | 48.7 | 39.1 | 37.4 | 37.7 | 25.3 | 26.2 |
| **CHAMP-Agg** | 34.8 | 34.9 | 36.5 | 31.1 | 31.1 | 37.8 | 41.3 | 33.5 | 32.4 | 41.8 | 45.2 | 37.2 | 34.9 | 36.1 | 24.4 | 24.1 |
| D3DP-Best* (Shan et al., 2023) | 35.4 | 33.0 | 34.8 | 31.7 | 33.1 | 37.5 | 43.7 | 34.8 | 33.6 | 45.7 | 47.8 | 37.0 | 35.0 | 35.0 | 24.3 | 24.1 |
| **CHAMP-Best*** | 32.8 | 32.9 | 33.2 | 28.9 | 29.6 | 35.7 | 41.4 | 32.3 | 31.4 | 39.7 | 43.9 | 36.1 | 32.9 | 32.8 | 22.8 | 22.8 |

Table 1: MPJPE ($\downarrow$) results on Human-3.6M dataset. Red: lowest error. Blue: second lowest error. In the single-hypothesis setting, our method runs without CP. In the multi-hypothesis setting, we compare four different variants of CHAMP as discussed in the main text. *Upper-bound performance, uses ground truth.

In the multi-hypothesis setting, we again set $H = 80$. CHAMP-Naive again yields fairly competitive results and when combined with conformal prediction and mean, J-Agg, and J-Best aggregation techniques, the performance continuously gets improved across the three metrics, with the CHAMP-Agg metric being the most competitive one, achieving SOTA results. These results here on this dataset follow the same trend as those on the Human3.6M dataset.

| | PCK $\uparrow$ | AUC $\uparrow$ | MPJPE $\downarrow$ |
|---|---|---|---|
| **Single Hypothesis** | | | |
| P-STMO (Shan et al., 2022) | 97.9 | 75.8 | 32.2 |
| MixSTE (Zhang et al., 2022b) | 96.9 | 75.8 | 35.4 |
| D3DP (Shan et al., 2023) | 97.7 | 77.8 | 30.2 |
| **CHAMP-Naive** | 97.9 | 76.0 | 30.1 |
| **Multiple Hypotheses** | | | |
| MHFormer (Li et al., 2022b) | 93.8 | 63.3 | 58.0 |
| DiffPose (Gong et al., 2023) | 98.0 | 75.9 | 29.1 |
| DiffPose (Feng et al., 2023) | 94.6 | 62.8 | 64.6 |
| D3DP-Agg (Shan et al., 2021) | 97.7 | 78.2 | 29.7 |
| **CHAMP-Naive** | 97.5 | 78.1 | 29.9 |
| **CHAMP-Naive-Agg** | 97.6 | 78.2 | 29.6 |
| **CHAMP** | 97.9 | 78.4 | 29.1 |
| **CHAMP-Agg** | 98.1 | 78.7 | 28.6 |
| D3DP-Best (Shan et al., 2021) | 98.0 | 79.1 | 28.1 |
| **CHAMP-Best** | 98.2 | 79.2 | 28.0 |

Table 2: Results on the 3DHP dataset.

## 5.3 In-the-Wild Videos

To show the generalizability of our method to in-the-wild videos, we collect videos from YouTube and Tik-Tok. To construct 2D keypoints input, we use Cascaded Pyramid Network (Chen et al., 2018) fine-tuned on the Human3.6M dataset as the default weights are trained on MSCOCO (Lin et al., 2014), which has a slightly different skeleton structure. We directly apply the CHAMP model trained on the Human3.6M dataset to test on in-the-wild videos. Sample results are shown in Fig. 1, where the input videos are collected from TikTok. Qualitative results suggest that CHAMP is able to filter out outlier hypotheses using the conformity score function trained end-to-end with the pose estimation model. More results are shown in Appendix G. For 3D visualization, please refer to this anonymized website to interact with CHAMP's predictions in 3D.

## 5.4 Ablation Studies with Human3.6M Dataset

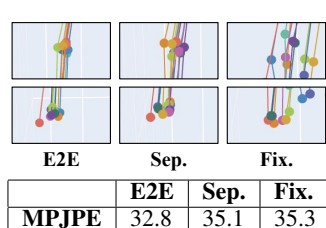

| | E2E | Sep. | Fix. |
|---|---|---|---|
| **MPJPE** | 32.8 | 35.1 | 35.3 |

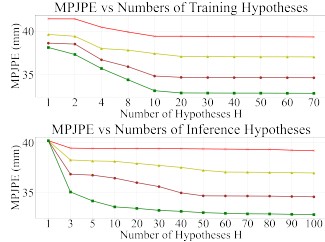

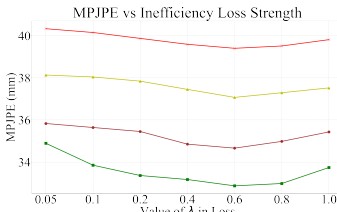

Figure 4: Comparison of conformity scores. **Top:** filtered hypotheses of two joints using the three scoring functions. **Bottom:** MPJPE (mm) values.

Figure 5: Comparison of #hypotheses in training and inference with 4 variants of CHAMP. Red: Naive, Yellow: CHAMP, Brown: Agg, Green: Best.

Figure 6: Comparison of $\lambda$ in loss definition $\mathcal{L}$ (Eq. 13) with 4 variants of CHAMP. Red: Naive, Yellow: CHAMP, Brown: Agg, Green: Best.

**Choice of the Conformity Function.** We compare our proposed end-to-end learned conformity score function with a separately trained conformity score function (Corso et al., 2022), and a hand-designed conformity score function. We compare the average performance across all actions on the Human3.6M dataset. For the separately trained conformity score function, we first only train the pose estimation backbone and generate 100 hypotheses per input sequence over a small subset of the training dataset. We then train the score function to predict if the proposed hypothesis conditioned on the input 2D keypoints results in an MPJPE of less than 25mm and use the ground-truth pose sequence as supervision. For the hand-designed conformity score, we use $\phi_{\text{peak}}$ from (Yang & Pavone, 2023), measuring the maximum MPJPE with respect to the ground truth across output frames. Note $\phi_{\text{peak}}$ measures "nonconformity", so we negate such values to fit our setup. All three variants use the J-Best aggregation method. From Fig. 4, end-to-end (E2E) score function yields the best performance and the two learned variants supersede the hand-designed version. In the two examples shown in Fig. 4, the filtered hypotheses seem to be more concentrated for learned score functions. It's worth noting that all three variants result in competitive performance, demonstrating the importance of CP.

**Number of Hypotheses.** Another interesting ablation study to conduct is to ablate on the number of hypotheses during training ($H_{\text{train}}$) and inference ($H$). We compare the effects of $H_{\text{train}}$, $H$ in Fig. 5, on the four variants of CHAMP. To compare the effect of $H_{\text{train}}$, we fix $H = 80$ and train a model for each value of hypotheses proposed during differentiable CP. Similarly, to compare the effect of $H$, we fix $H_{\text{train}} = 20$ during training and use the same model with different $H$ during inference for CP. Results suggest that a good performance can be achieved with $H_{\text{train}} = 20$ during training, while any number higher than 20 brings marginal improvement and requires more GPU memory. Moreover, $H = 80$ during inference is sufficient for all four variants.

**Strength of the Inefficiency Loss.** Finally, we ablate on the value of $\lambda$ in the overall training objective $\mathcal{L}$. This is an important ablation in that we can find suitable strength of the inefficiency loss $\mathcal{L}_{\text{ineff}}$ to make sure it does not conflict $\mathcal{L}_{\text{pose}}$. Results in Fig. 6 suggest that $\lambda = 0.6$ is the most efficient strength across all values, as smaller $\lambda$ does not train the scoring model sufficiently and higher $\lambda$ conflicts with the pose loss. This corroborates the smaller-scale hyperparameter sweep experiments we conducted before training the models.

**Correlation between Conformity Score and Prediction Error.** We also investigate if the learned conformity score is correlated with prediction error (MPJPE). To achieve this, we collect 500 random test samples and generate hypotheses as well as the predicted conformity score values. We plot these values in Fig. 7. Experiment data suggest that the predicted conformity score has a Pearson correlation value of -0.41 with the MPJPE value. We further fit an Ordinary Least Squares (OLS) regression line by regressing the predicted score onto the prediction error. The resulting OLS fit achieves an $R^2$ of 0.26, demonstrating that the conformity score has a fairly strong explanatory power to explain the prediction error.

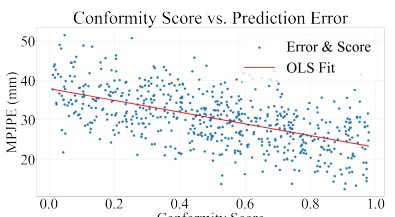

Figure 7: Correlation and OLS fit between MPJPE and predicted conformity score.

**Other Uncertainty-Augmented Human Pose Estimation Methods.** We now compare different implementations for uncertainty estimation in human pose estimation. Several prior works have explored uncertainty in 3D human pose/shape reconstruction (Rempe et al., 2021; Dwivedi et al., 2024; Biggs et al., 2020; Zhang et al., 2023; 2024). The cited works all simultaneously recover shape and pose and we only compare with the pose metrics. While the methods all incorporate uncertainty in the pose estimation process, sampling with the (learned) uncertainty remains nontrivial due to the lack of flexibility. Moreover, multiple-hypothesis-based methods from probabilistic generative models (Shan et al., 2023; Gong et al., 2023) significantly outperform other models. CHAMP builds on top of this and added an extra layer of uncertainty

| Baseline | MPJPE (mm) |
|---|---|
| 3DMB (Biggs et al., 2020) | 58.2 |
| HuMOR (Rempe et al., 2021) | 49.9 |
| POCO (Dwivedi et al., 2024) | 47.7 |
| RoHM (Zhang et al., 2024) | 41.6 |
| KNOWN (Zhang et al., 2023) | 40.5 |
| D3DP Shan et al. (2023) | 39.5 |
| DiffPose Gong et al. (2023) | 36.9 |
| **CHAMP (Ours)** | 34.8 |

Table 3: Comparison with other uncertainty design choices.

learning, which further improves the generative model's performance. We also provide detailed design choice comparisons with the aforementioned methods in the Appendix F.

## 5.5 EXCHANGEABILITY AND CP GUARANTEE

The inherent assumption of CP requires the calibration dataset to be exchangeable, which is weaker than asking them to be independent (Angelopoulos & Bates, 2021). This assumption typically fails when the calibration set is a single video sequence, where the image frames are temporally correlated. We learn a sequence-to-sequence model, where the input and output are long sequences of image frames from different videos, instead of single frames from the same video. This would drastically decrease the temporal correlation across the calibration dataset. Moreover, as shown in (Yang & Pavone, 2023), the video frames tend to be more independent if the videos are taken by multiple evenly spaced cameras, which is the case for our datasets of interest. We also investigate the empirical coverage of our method in Human 3.6M test set in Appendix A. Results suggest that we are able to inherit the coverage guarantee from CP with the learned conformity score even if the dataset is not fully exchangeable.

## 5.6 IMPLEMENTATION AND TRAINING DETAILS

CHAMP's denoiser model uses Adam optimizer with a weight decay parameter of 0.1 and momentum parameters $\beta_1 = \beta_2 = 0.99$. For the training objective in eq. (13), we use $\lambda = 0.6$. We train the CHAMP model using an NVIDIA V100 GPU, where the training process consumes an amortized GPU memory of 30GB, for 300 epochs with a batch size of 8 and a learning rate of 5e-5 and reduce it on plateau with a factor of 0.5. Following previous work (Zhang et al., 2022b; Shan et al., 2021; 2023), we use input pose sequence of 243 frames ($N = 243$) of Human3.6M universal skeleton format. During training, the number of hypotheses is 20, and #DDIM iterations is set to 1. During inference, they are set to 80 and 10. The maximum number of diffusion steps is $T = 999$.

## 6 LIMITATIONS

While our method shows a promising improvement over the current 3D human pose estimation methods, it does have several limitations. First, many hypotheses need to be proposed during training to improve the learned conformity score, which consumes a lot more GPU memory. Second, a lot of computation is needed for training the scoring function, making the whole training process fairly slow. Lastly, CHAMP only learns single-human 3D skeleton estimation, without considering human shape estimation.

## 7 CONCLUSION

This work presents CHAMP, a novel method for learning multi-hypotheses 3D human poses with a learned conformal predictor from 2D keypoints. We empirically show that CHAMP achieves competitive results across multiple metrics and datasets, and when combined with more sophisticated downstream aggregation methods, it achieves state-of-the-art performance. Future work includes using more recent CP techniques that relax the exchangeability assumption (Barber et al., 2023), using more efficient sequence-to-sequence models such as Mamba (Gu & Dao, 2023), and scaling the pipeline up to dense human pose-shape joint prediction scenarios (Loper et al., 2023).

## 8 ETHICS STATEMENT

We take ethics very seriously and our research conforms to the ICLR Code of Ethics. Human pose estimation is a well-established research area, and this paper inherits all the impacts of the research area, including potential for dual use of the technology in both civilian and military applications. We believe that the work does not impose a high risk for misuse. Furthermore, the paper does not involve crowdsourcing or research with human subjects.

## 9 Reproducibility Statement

Our paper makes use of publicly available open-source datasets, ensuring that the data required for reproducing our results is accessible to all researchers. We have thoroughly documented all aspects of our model's training, including the architecture, hyperparameters, optimizer settings, learning rate schedules, and any other implementation details for achieving the reported results. These details are provided in Section 5 and Appendix C, D of our paper. Additionally, we specify the hardware and software configurations used for our experiments to facilitate replication. We anticipate that it should not be challenging for other researchers to reproduce the results and findings presented in this paper.

## Acknowledgements

This work was partially funded by Amazon Robotics under the "Safe Autonomy Leveraging Intelligent Optimization" program, by the ARL DCIST program, and by the ONR RAPID program.

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

## A    COVERAGE GUARANTEE AND EMPIRICAL COVERAGE

Please refer to the standard proof shown in (Angelopoulos & Bates, 2021; Shafer & Vovk, 2008) for the coverage guarantees of conformal prediction.

To evaluate the empirical coverage of CHAMP, we calculate the mean coverage percentage across test data. Specifically, we use eq. (14) to check if the ground truth belongs in the confidence set formed by the conformity function on the test data:

$$\bar{\mathcal{C}} = \frac{1}{|I_{\text{test}}|} \sum_{\boldsymbol{y}_{\text{GT}} \sim I_{\text{test}}} \mathbb{1}\left(\boldsymbol{y}_{\text{GT}} \in C_\theta(\boldsymbol{x}, \tau)\right)$$

$$= \frac{1}{|I_{\text{test}}|} \sum_{\boldsymbol{y}_{\text{GT}} \sim I_{\text{test}}} \mathbb{1}\left(\phi_\theta(\boldsymbol{y}_{\text{GT}}) \geq \tau\right)$$

We compare the three choices of conformity scores for the empirical coverage calculation. We use the test set from Human 3.6M and calculate the empirical coverage values across all activities. During training and testing, we keep $\alpha = 0.1$ for CP. From Fig. 8, we see the empirical coverage is

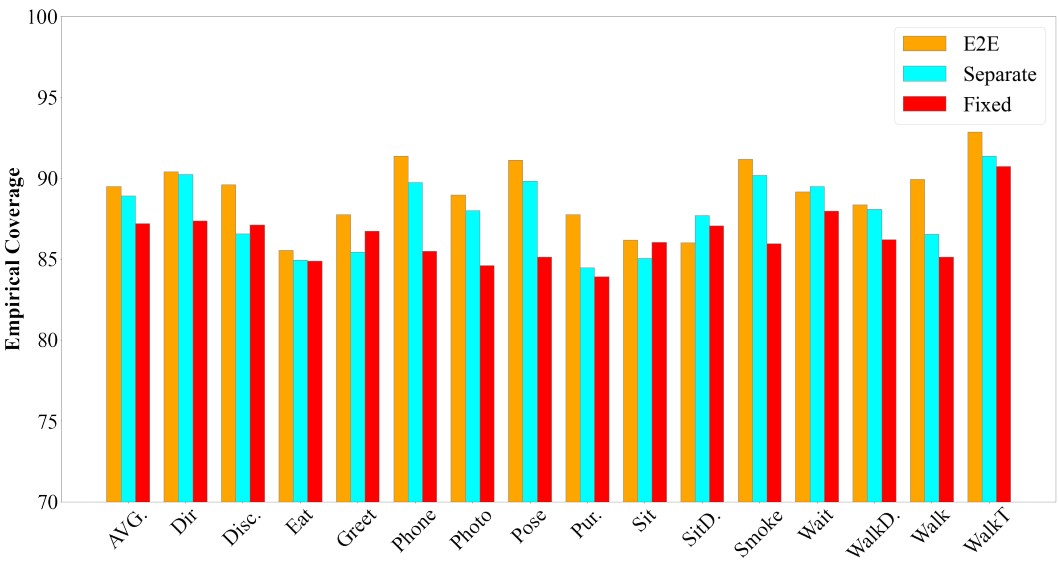

Figure 8: Empirical coverage comparison across three different conformity functions. We keep the best $\lambda = 0.6$ for comparison.

around $90\% \pm 5\%$ for all activities, which remains close to $1 - \alpha$, and in some cases, it exceeds this value. This is encouraging given the exchangeability assumption is not fully satisfied in our dataset.

## B    DENOISER IN TRAINING VS INFERENCE

We first revisit DDIM (Song & Ermon, 2020). We start with the reverse process derivation and rewrite $q(\boldsymbol{y}_{t-1}|\boldsymbol{y}_t, \boldsymbol{y}_0)$ to be parameterized by a desired standard deviation $\sigma_t$:

$$\boldsymbol{y}_{t-1} = \sqrt{\alpha_{t-1}}\boldsymbol{y}_0 + \sqrt{1 - \alpha_{t-1}}\boldsymbol{\epsilon}_{t-1}$$

$$= \sqrt{\alpha_{t-1}}\boldsymbol{y}_0 + \sqrt{1 - \alpha_{t-1} - \sigma_t^2}\boldsymbol{\epsilon}_{t-1} + \sigma_t\boldsymbol{\epsilon}$$

$$= \sqrt{\alpha_{t-1}}\left(\boldsymbol{y}_t - \sqrt{1 - \alpha_t}\boldsymbol{\epsilon}_\theta^{(t)}(\boldsymbol{y}_t)\right) + \sqrt{1 - \alpha_{t-1} - \sigma_t^2}\boldsymbol{\epsilon}_\theta^{(t)}(\boldsymbol{y}_t) + \sigma_t\boldsymbol{\epsilon}$$

$$= \sqrt{\alpha_{t-1}}\left(\boldsymbol{y}_t - \sqrt{1 - \alpha_t}\boldsymbol{\epsilon}_\theta^{(t)}(\boldsymbol{y}_t)\right) + \sqrt{1 - \alpha_{t-1} - \sigma_t^2}\boldsymbol{\epsilon}_\theta^{(t)}(\boldsymbol{y}_t) + \sigma_t\boldsymbol{\epsilon}$$

where the model $\boldsymbol{\epsilon}_\theta^{(t)}(\cdot)$ predicts the $\boldsymbol{\epsilon}_t$ from $\boldsymbol{y}_t$. Since $q(\boldsymbol{y}_{t-1}|\boldsymbol{y}_t, \boldsymbol{y}_0) = \mathcal{N}(\boldsymbol{y}_{t-1}; \mu(\boldsymbol{y}_t, \boldsymbol{y}_0), \beta_t \mathbf{I})$, therefore we have:

$$\tilde{\beta}_t = \sigma_t^2 = \frac{1 - \bar{\alpha}_{t-1}}{1 - \bar{\alpha}_t} \beta_t$$

Let $\sigma_t^2 = \eta \cdot \beta_t$ where $\eta$ is a hyperparameter that controls the sampling stochasticity. During generation, we don't have to follow the whole chain $t = 1, \ldots, T$, but only a subset of steps. Denote $t' < t$ as two steps in this accelerated trajectory. The DDIM update step is defined as follows:

$$q_{t',\theta}(\boldsymbol{y}_{t'}|\boldsymbol{y}_t, \boldsymbol{y}_0) = \mathcal{N}\left(\boldsymbol{x}_{t'}; \sqrt{\alpha_{t'}}\left(\boldsymbol{y}_t - \sqrt{1 - \alpha_t}\boldsymbol{\epsilon}_\theta^{(t)}(\boldsymbol{y}_t)\right), \sigma_t^2 \mathbf{I}\right)$$

Following D3DP (Shan et al., 2023), the denoiser model $D_\theta$ uses DDIM to sample denoised poses from the corrupted ones. During training, we run DDIM for only 1 step for the sake of efficiency:

$$\bar{\boldsymbol{y}}^h = D_\theta(\boldsymbol{y}_T^h, \boldsymbol{x}, T), \; \boldsymbol{y}_T^h \sim \mathcal{N}(0, \mathbf{I}) \quad \forall h = \{1, \cdots, H\}$$

During inference, we run DDIM for $K = 10$ times, and each step is defined as:

$$t = T \cdot (1 - k/K), \; t' = T \cdot (1 - (k+1)/K)), \; k = \{0, \cdots, K-1\}$$
$$\bar{\boldsymbol{y}}^h = D_\theta(\bar{\boldsymbol{y}}_t^h, \boldsymbol{x}, t), \quad \forall h = \{1, \cdots, H\}$$
$$\bar{\boldsymbol{y}}_{t'}^h = \sqrt{\bar{\alpha}_{t'}} \cdot \bar{\boldsymbol{y}}^h + \sqrt{1 - \bar{\alpha}_{t'} - \sigma_t^2} \cdot \boldsymbol{\epsilon}_t + \sigma_t \boldsymbol{\epsilon}$$
$$t \leftarrow t'$$

## C  TRAINING-CALIBRATION-TESTING DATA SPLIT

We discuss the split of training, calibration, and testing data. We first split the testing data the same way as (Zhang et al., 2022b; Shan et al., 2023; Gong et al., 2023) to ensure the fairness of results. We then further split the training set into the actual training dataset and a calibration dataset before inference. Specifically, we split the training dataset by uniformly sampling a 2% subset as the calibration dataset, and CHAMP is only trained on the remaining 98%. Note here that this 2% calibration dataset is not seen during training by any means. During training, we split each mini-batch evenly into $B_{\text{pred}}$ and $B_{\text{cal}}$.

## D  ARCHITECTURE DETAILS

We provide a more detailed version of Fig. 2. We illustrate the detailed architecture of the denoiser model $D_\theta$ as well as the conformity scoring model $s_\theta$ in Fig. 9. MixSTE (Zhang et al., 2022b) is used as the backbone of the denoiser. MixSTE combines 16 alternative spatial and temporal attention blocks. In the implementation, the channel size is set to 512. Similar to (Shan et al., 2023; Gong et al., 2023), we concatenate 2D keypoints and noisy 3D poses as inputs and add a diffusion timestep embedding after the input embedding using a sinusoidal function. For the conformity scoring model, we concatenate 2D keypoints and predicted hypotheses 3D poses as inputs, reuse the input embedding layer, and use an extra MLP to project the input embeddings into a score in the range [0, 1]. In training, CP is made differentiable by using soft ranking and soft assignment, resulting in a differentiable inefficiency loss $\mathcal{L}_{\text{ineff}}$. This loss is combined with the pose loss $\mathcal{L}_{\text{pose}}$ when backpropagating into the network. During inference (light cyan shaded area), we refine the hypotheses confidence set with regular CP on a held-out calibration set, resulting in $H' < H$ hypotheses. We then aggregate the refined set, resulting in a single final prediction.

## E  PROCRUSTES-MPJPE PERFORMANCE

The Procrustes MPJPE (P-MPJPE) metric, which is often referred to as Protocol #2, computes MPJPE after the estimated poses align to the ground truth using a rigid transformation. It is another

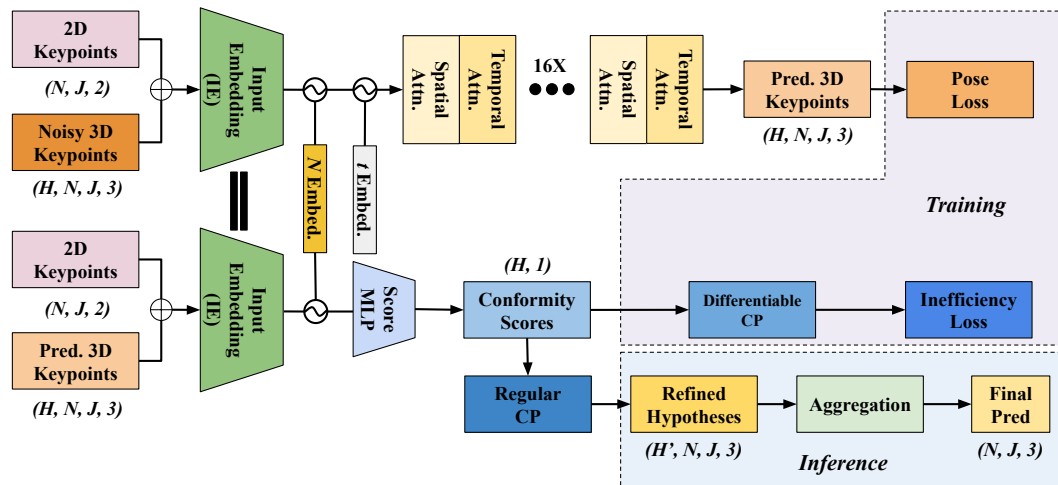

Figure 9: Detailed architecture of $D_\theta$ and $S_\theta$, the denoiser and conformity score models during training and inference. During training, the construction of refined hypotheses set after differentiable CP is done via soft assignment. During inference, regular CP is used with the trained conformity score function.

standard metric for measuring a 3D pose estimator's performance (Pan et al., 2023). We report the performance of the four variants of CHAMP against other baselines on the Human3.6M dataset in Table 4. Quantitative results suggest our method achieves SOTA performance on this metric as well.

| Procrustes Per-Joint Position Error (Protocol # 2) - mm | | | | | | | | | | | | | | |
|---|---|---|---|---|---|---|---|---|---|---|---|---|---|---|
| | AVG. | Dir | Disc. | Eat | Greet | Phone | Photo | Pose | Pur. | Sit | SitD. | Smoke | Wait | WalkD. | Walk | WalkT. |
| **Single Hypothesis** | | | | | | | | | | | | | | | | |
| STE Li et al. (2022a) | 35.2 | 32.7 | 35.5 | 32.5 | 35.4 | 35.9 | 41.6 | 33.0 | 31.9 | 45.1 | 50.1 | 36.4 | 33.5 | 35.1 | 23.9 | 25.0 |
| P-STMO Shan et al. (2022) | 34.4 | 31.3 | 35.2 | 32.9 | 33.9 | 35.4 | 39.3 | 32.5 | 31.5 | 44.6 | 48.2 | 36.3 | 32.9 | 34.4 | 23.8 | 23.9 |
| MixSTE Zhang et al. (2022b) | 32.6 | 30.8 | 33.1 | 30.3 | 31.8 | 33.1 | 39.1 | 31.1 | 30.5 | 42.5 | 44.5 | 34.0 | 30.8 | 32.7 | 22.1 | 22.9 |
| DUE Zhang et al. (2022a) | 32.5 | 30.3 | 34.6 | 29.6 | 31.7 | 31.6 | 38.9 | 31.8 | 31.9 | 39.2 | 42.8 | 32.1 | 32.6 | 31.4 | 25.1 | 23.8 |
| D3DP Shan et al. (2023) | 31.7 | 30.6 | 32.5 | 29.1 | 31.0 | 31.9 | 37.6 | 30.3 | 29.4 | 40.6 | 43.6 | 33.3 | 30.5 | 31.4 | 21.5 | 22.4 |
| **CHAMP-Naive** | 31.0 | 30.7 | 32.0 | 29.1 | 30.6 | 31.1 | 35.8 | 29.1 | 28.4 | 39.7 | 41.4 | 33.0 | 30.1 | 31.3 | 20.8 | 22.3 |
| **Multiple Hypotheses** | | | | | | | | | | | | | | | | |
| MHFormer Li et al. (2022b) | 34.4 | 34.9 | 32.8 | 33.6 | 35.3 | 39.6 | 32.0 | 32.2 | 43.5 | 48.7 | 36.4 | 32.6 | 34.3 | 23.9 | 25.1 | 34.4 |
| GraphMDN Oikarinen et al. (2021) | 46.9 | 39.7 | 43.4 | 44.0 | 46.2 | 48.8 | 54.5 | 39.4 | 41.1 | 55.0 | 69.0 | 49.0 | 43.7 | 49.6 | 38.4 | 42.4 |
| DiffuPose Choi et al. (2023) | 39.9 | 35.9 | 40.3 | 36.7 | 41.4 | 39.8 | 43.4 | 37.1 | 35.5 | 46.2 | 59.7 | 39.9 | 38.0 | 41.9 | 32.9 | 34.2 |
| DiffPose Gong et al. (2023) | 28.7 | 26.3 | 29.0 | 26.1 | 27.8 | 28.4 | 34.6 | 26.9 | 26.5 | 36.8 | 39.2 | 29.4 | 26.8 | 28.4 | 18.6 | 19.2 |
| DiffPose Feng et al. (2023) | 32.0 | 28.1 | 31.5 | 28.0 | 30.8 | 33.6 | 35.3 | 28.5 | 27.6 | 40.8 | 44.6 | 31.8 | 32.1 | 32.6 | 28.1 | 26.8 |
| D3DP-Agg Shan et al. (2023) | 31.6 | 30.6 | 32.4 | 29.2 | 30.9 | 31.9 | 37.4 | 30.2 | 29.3 | 40.4 | 43.2 | 33.2 | 30.4 | 31.3 | 21.5 | 22.3 |
| **CHAMP-Naive** | 31.2 | 30.4 | 32.9 | 30.4 | 31.0 | 32.0 | 37.1 | 28.6 | 28.1 | 40.2 | 42.4 | 33.9 | 30.1 | 30.1 | 21.2 | 22.1 |
| **CHAMP** | 28.7 | 26.8 | 29.4 | 26.7 | 28.8 | 29.1 | 36.1 | 25.8 | 24.9 | 37.5 | 39.7 | 30.9 | 29.1 | 27.6 | 19.0 | 20.0 |
| **CHAMP-Agg** | 27.9 | 26.1 | 28.7 | 25.6 | 28.1 | 28.2 | 35.1 | 25.3 | 24.4 | 37.1 | 38.4 | 30.1 | 28.8 | 25.9 | 18.2 | 19.4 |
| D3DP-Best* Shan et al. (2023) | 28.7 | 27.5 | 29.4 | 26.6 | 27.7 | 29.2 | 34.3 | 27.5 | 26.2 | 37.3 | 39.0 | 30.3 | 27.7 | 28.2 | 19.6 | 20.3 |
| **CHAMP-Best*** | 27.1 | 25.9 | 28.1 | 25.2 | 27.3 | 28.1 | 34.4 | 24.7 | 23.1 | 35.2 | 37.9 | 28.9 | 27.4 | 23.9 | 18.1 | 19.3 |

Table 4: P-MPJPE ($\downarrow$) results on Human-3.6M dataset. *Upper-bound performance, needs ground truth.

## F  COMPARISON TO OTHER UNCERTAINTY DESIGN CHOICES

We now discuss different design choices for uncertainty estimation in human pose estimation. Several prior works have explored uncertainty in 3D human pose/shape reconstruction (Rempe et al., 2021; Dwivedi et al., 2024; Biggs et al., 2020; Zhang et al., 2023; 2024). While methods such as (Dwivedi et al., 2024) also learn an explicit confidence value for occlusion, using the confidence to sample good poses is non-trivial. Moreover, motion-based methods such as (Zhang et al., 2024; Rempe et al., 2021) use physical contacts and trajectory consistency to make the uncertain estimates more robust, which does not explicitly model the "quality" of the samples. Zhang et al. (2023) use explicit anatomy constraints and also shows that uncertainty modeling can indeed improve model performance. Lastly, Biggs et al. (2020) generates a fixed number of hypotheses and learns to choose the best ones, which is similar in flavor to CHAMP but lacks the probabilistic flexibility and the coverage guarantee from CP. While all aforementioned works are valid design choices for uncertainty in human pose learning, CHAMP lends itself particularly well to probabilistic pose estimation meth-

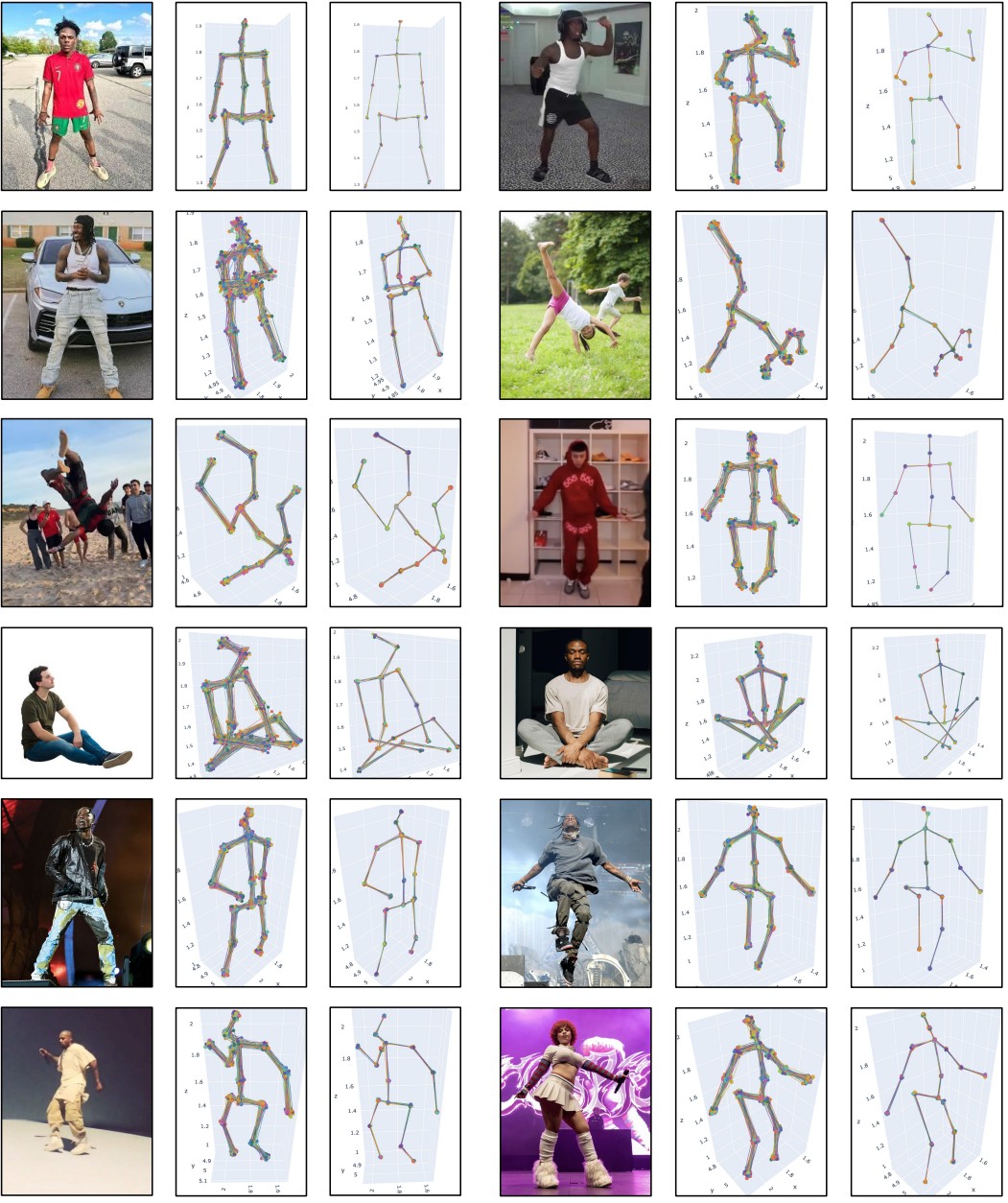

Figure 10: Results of running CHAMP on in-the-wild videos collected from TikTok and YouTube.

ods, which can then be augmented with an explicit uncertainty modeling function learned in tandem with the pose estimation model.

## G    MORE QUALITATIVE RESULTS ON IN-THE-WILD VIDEOS

We showcase the performance of CHAMP on more in-the-wild videos in this section. We collect various YouTube and TikTok videos and detect 2D keypoints with Cascaded Pyramid Network. Note that the official weights provided by CPN are trained in COCO format, so we used a fine-tuned version of the weights for Human3.6M universal skeleton format. With such 2D keypoints, we are able to directly use the model trained on Human3.6M dataset and apply it to real-world videos without any fine-tuning. In Fig. 10, for each row, we show two examples. For every three images,

the leftmost image is the RGB observation, and the middle image is all the hypotheses proposed by CHAMP's pose estimation backbone, and the rightmost image is the hypotheses set refined by the learned conformal predictor.

# H    LEARNED CP IN CHAMP

We provide more demonstrations of the learned conformal predictor in CHAMP. Specifically, in Fig. 11, we provide more examples of predicted hypotheses before and after the conformal predictor powered by the learned score function.

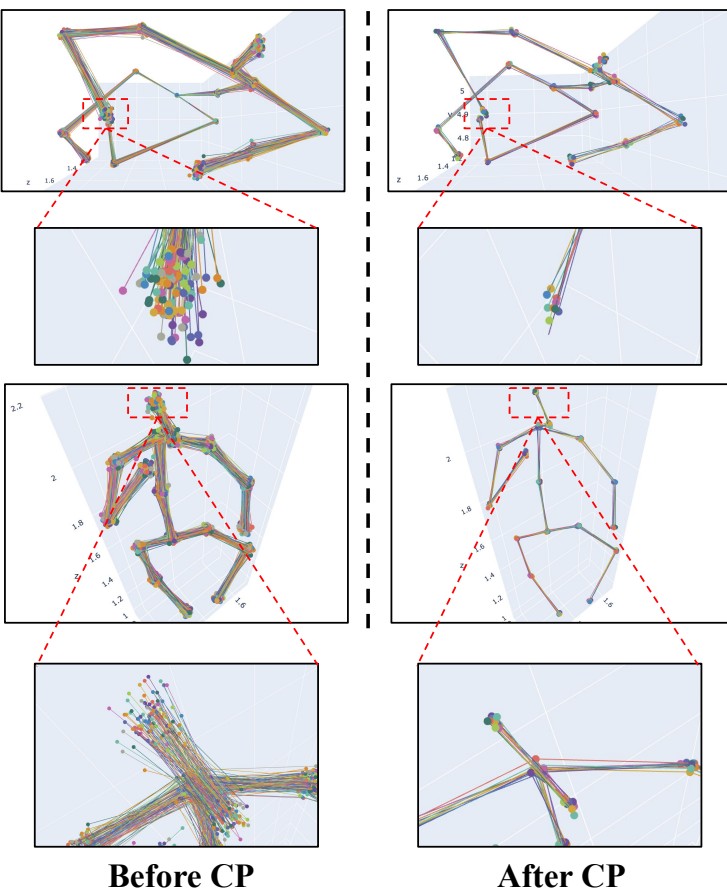

Figure 11: Examples of CP in CHAMP filtering out bad hypotheses during inference. Qualitatively, the learned score function filters out outlier predictions.

# I    CONFORMITY SCORE VISUALIZATION

We also show some qualitative visualization of the learned conformity score. We use heatmap to color-code different hypotheses.

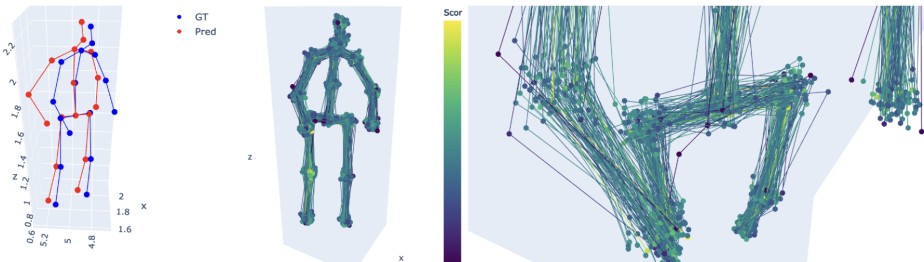

Figure 12: Failure case and the predicted conformity score. While we do see that hypotheses further away from the ground truth get lower weight, most of the hypotheses here are off, significantly skewing the final output. Moreover, we see that the variance of the estimate of the most incorrect joint (i.e. the right elbow) does appear a lot higher than others, looking more spread out. This shows that higher uncertainty tends to be associated with higher error.

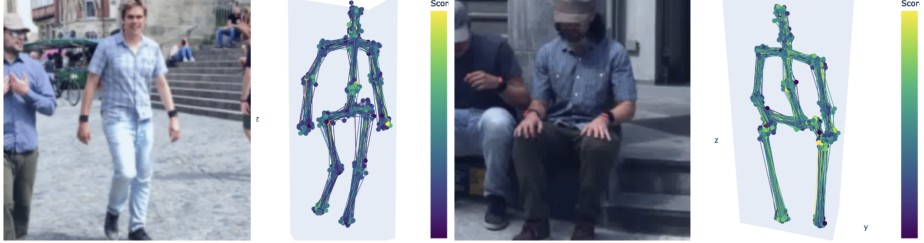

Figure 13: Learned conformity score prediction on 3DPW data. We see that in both cases, the conformity score is higher in the more concentrated areas and vice versa.

## J  SINGLE-FRAME VARIATION

When demonstrating the effectiveness of the proposed method, one problem is that the ambiguity in video 3D pose estimation where temporal information is abundant is not as significant as in the single-frame case. Thus, we are interested in if the proposed CP pipeline could be beneficial for single-frame image-based 3D pose estimation. Note that this requires minimal change to our method pipeline as we can adjust the number of frames ($N$) to 1 from 243. We thus train our method as well as the baseline methods with $N = 1$ and test on the same dataset.

| Single Frame ($N = 1$) Mean Per-Joint Position Error (Protocol # 1) - mm | | | | | | | | | | | | | | | | |
|---|---|---|---|---|---|---|---|---|---|---|---|---|---|---|---|---|
| | AVG. | Dir | Disc. | Eat | Greet | Phone | Photo | Pose | Pur. | Sit | SitD. | Smoke | Wait | WalkD. | Walk | WalkT. |
| DiffPose (Gong et al., 2023) | 40.5 | 37.5 | 39.2 | 40.1 | 40.7 | 40.1 | 48.2 | 39.0 | 35.5 | 53.1 | 55.1 | 41.1 | 39.6 | 39.7 | 28.7 | 28.9 |
| D3DP (Shan et al., 2023) | 42.1 | 40.1 | 41.2 | 41.1 | 42.1 | 44.3 | 50.2 | 41.8 | 39.4 | 51.2 | 56.7 | 43.2 | 40.4 | 40.3 | 30.5 | 29.1 |
| **CHAMP-Naive** | 41.9 | 39.8 | 41.1 | 40.8 | 41.8 | 43.9 | 49.1 | 40.9 | 38.6 | 51.1 | 56.4 | 43.1 | 39.6 | 39.8 | 30.1 | 28.9 |
| **CHAMP-Naive-Agg** | 40.7 | 38.9 | 39.9 | 39.4 | 40.4 | 42.8 | 48.9 | 39.8 | 38.1 | 50.8 | 55.2 | 41.2 | 39.1 | 38.8 | 29.3 | 27.7 |
| **CHAMP** | 38.6 | 38.2 | 39.4 | 34.5 | 36.3 | 40.1 | 46.2 | 37.1 | 36.4 | 46.9 | 50.9 | 40.4 | 38.2 | 38.1 | 27.2 | 27.7 |
| **CHAMP-Agg** | 37.1 | 36.5 | 38.1 | 33.1 | 34.2 | 38.4 | 44.1 | 35.1 | 33.9 | 44.9 | 49.2 | 40.2 | 37.9 | 37.8 | 26.5 | 27.2 |

Table 5: MPJPE ($\downarrow$) results on Human-3.6M dataset. Note that all baselines shown are trained with input frames number $N = 1$.

As expected, with fewer frames available, all tested baselines should result in worse performance than with multiple frames. Interestingly, as we can see in Table 5, CHAMP-Agg is able to achieve a more significant edge over the previous SOTA (DiffPose) in single-frame setting vs in multiframe (3.4mm vs 2.1mm). Moreover, CHAMP-Agg achieves more improvement over the backbone D3DP in the single-frame setting. Thus, even with less temporal information available (i.e. when the ambiguity is more severe), CHAMP achieves even more competitive performance compared to baselines.

## K  DIVERSE HYPOTHESES UNDER LARGE UNCERTAINTY

Another interesting study to conduct is to test the proposed method's behaviors under large uncertainty. Here, we specify large uncertainty as three scenarios:

- Truncation in 2D RGB input.
- Wrong 2D keypoints prediction.
- Self-occlusion in 2D RGB input.

While first two types large uncertainty, truncation in 2D RGB input (which gives inaccurate 2D keypoints) and wrong 2D keypoints, are not the focus of our work (since we assume 2D keypoints from some off-the-shelf method), it would be interesting to see how CHAMP behaves when such uncertainty occurs. To test this, we use the single-frame model trained in the previous section —this allows us to isolate the temporal effects and focus on uncertainty in 2D RGB inputs only, as the temporal nature of the data may already provide additional constraints, reducing ambiguity.

### K.1  TRUNCATED RGB INPUT

When the RGB image input is truncated, meaning that some keypoints are not fully visible, it might be possible that the 2D keypoints detector can still predict decent 2D keypoints. As shown in CPN Chen et al. (2018) and HRNet Sun et al. (2019), the 2D keypoints detector does have some robustness to truncation (external occlusion). We synthetically create truncated (externally occluded) images and test CHAMP on them, shown in Fig. 14. In the first case, the two feet are truncated and in the second case, one hand is truncated. In both cases, the 2D keypoints detector was able to predict the truncated 2D keypoints. Since our method takes as input 2D keypoints, so long as all keypoints are available, it will run as usual, agnostic to the truncation.

### K.2  WRONG 2D KEYPOINTS

It is also likely that the 2D keypoints detector makes mistakes due to noisy background, imprecise segmentation, etc. Under this circumstance, the 2D keypoints are off to begin with. Granted, when the 2D keypoints are extremely wrong, the 3D keypoints output will also be off —this constitutes

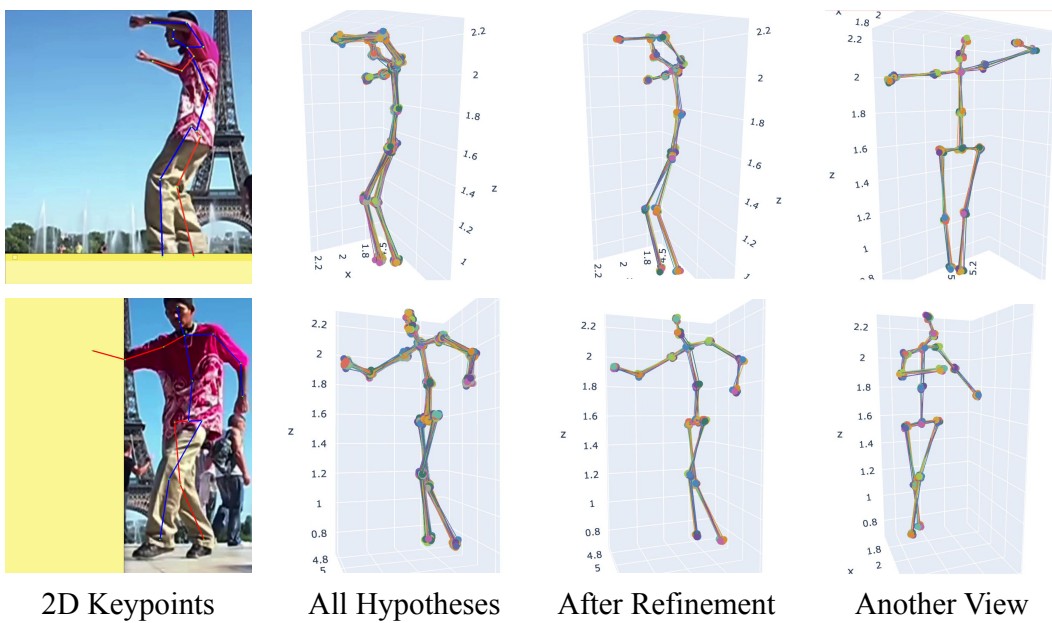

| 2D Keypoints | All Hypotheses | After Refinement | Another View |
|---|---|---|---|

Figure 14: CHAMP Performance on Truncated Images. As long as the keypoints are detected, the pipeline can output 3D keypoints as usual. This relies on 2D keypoint detector's robustness to external occlusion.

one key failure mode of CHAMP. However, we are interested to see if the uncertainty score in CHAMP is able to "correct" the output by proposing more diverse samples. In Fig. 15, we select two images where the 2D keypoint detector makes mistakes. In the first image, the left leg keypoints are completely off. Interestingly, the proposed 3D keypoints are a lot more diverse for the left leg than other joints. After the CP refinement step, the left leg joints are still more diverse, covering a wider region than other joints outputs do. In the second image, the two hands keypoints are off. We again see a much noisier proposal set for the two joints. After the refinement, the two hands outputs are noticeably more diverse. One possible reason for this interesting phenomenon could be that the 2D keypoints input itself are seen as "unnatural" compared to the training data, forcing the network to output more diverse 3D poses.

### K.3 SELF-OCCLUDED RGB INPUTS

Lastly, another source of large uncertainty in RGB inputs is self-occlusion. Note that self-occlusion typically happens when the human subject itself occludes some joints, and this is different from truncation, where the source of occlusion is external. We are interested in testing CHAMP when self-occlusion is present and checking if any interesting behavior occurs. In Fig. 16, we show two images where the human subject occludes itself, obstructing the visibility of some joints. In the first image, the dancer's left arm is occluded by the right arm entirely. While the 2D detector is able to differentiate the two joints and output two different 2D keypoints, they are very close and the 3D locations are ambiguous. CHAMP outputs noisier proposals for the occluded arm joints (elbow and hand). Interestingly, after the CP refinement, the elbow joint pose outputs are almost "bimodal" and the hand pose outputs are still a lot noisier than other joints. For the second image, the dancer's right leg is occluded by the left leg. Similarly, CHAMP outputs noisier proposals for the right knee and right foot both before and after the CP refinement. One possible reason for this interesting phenomenon is that the network intrinsically learned to output different poses when the 2D keypoints are close to each other, suggesting possible self-occlusion.

### L PERFORMANCE ON 3DPW

The 3DPW dataset Von Marcard et al. (2018) is a more challenging dataset collected in outdoor environment using IMU (inertial measurement unit) sensors with mobile phone lens. To ensure a

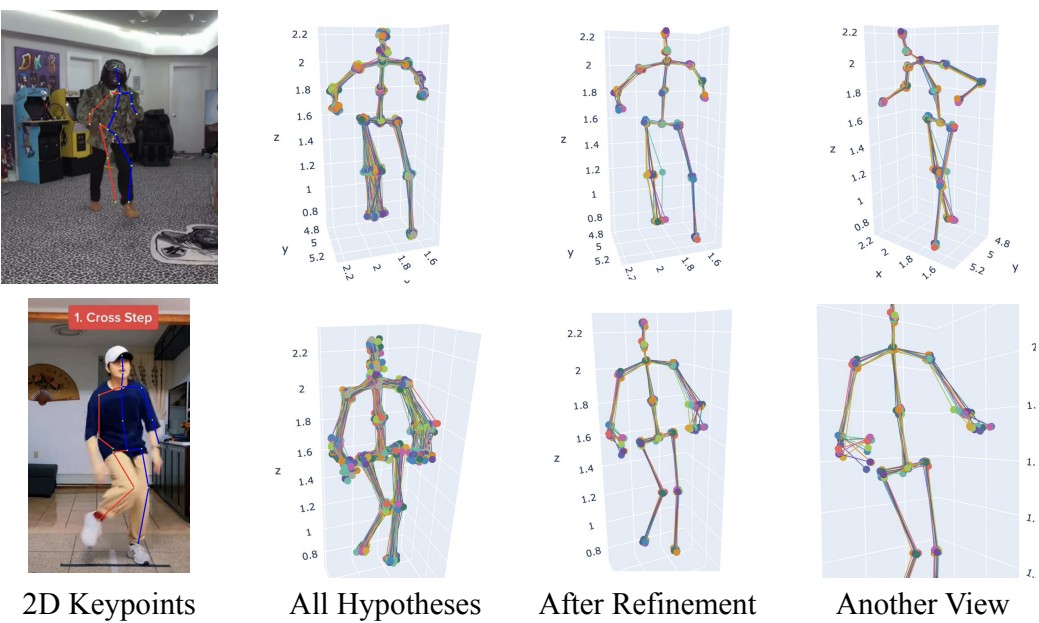

| 2D Keypoints | All Hypotheses | After Refinement | Another View |

Figure 15: CHAMP Performance on Wrong 2D Keypoints. Interestingly, CHAMP outputs a lot more diverse proposals for the joints that are off than for more correct joints.

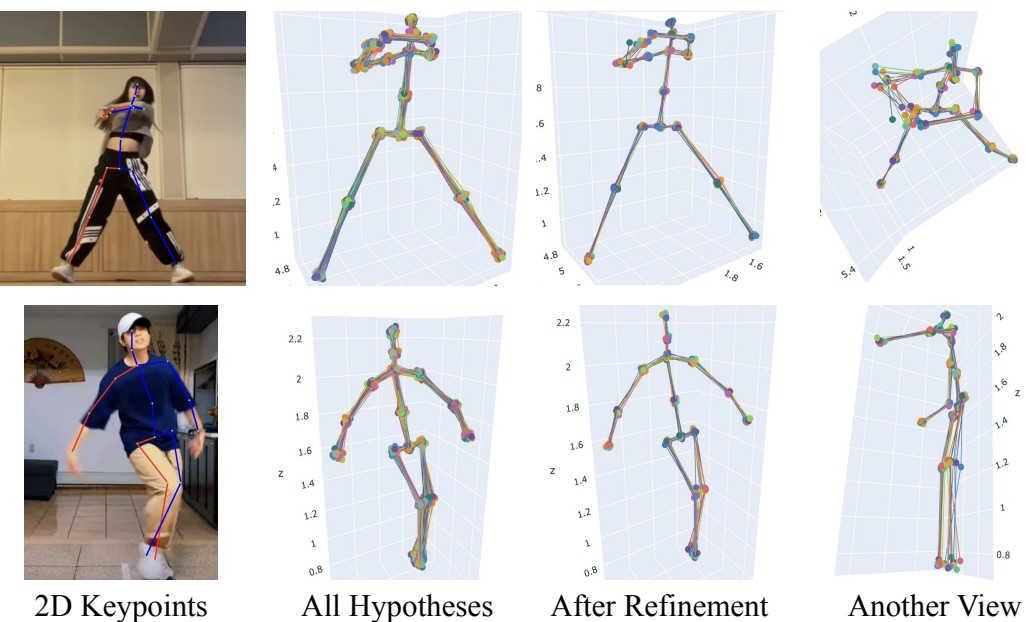

| 2D Keypoints | All Hypotheses | After Refinement | Another View |

Figure 16: CHAMP Performance on Self-Occluded RGB Inputs. Interestingly, CHAMP outputs a lot more diverse proposals for the joints that are occluded than for non-occluded joints.

fair comparison, following Yang et al. (2023), we compare CHAMP with other baselines by taking as input *ground-truth* 2D keypoints. We compare CHAMP ($N = 1$), D3DP ($N = 1$) Shan et al. (2023), DiffPose ($N = 1$) Gong et al. (2023), and CameraPose Yang et al. (2023) trained jointly on Human 3.6M and MPII without using 3DPW during training and test on 3DPW using the PA-MPJPE (aligned after rigid transformation) in Table 6. Note that in Shan et al. (2023) did not provide quantitative results on 3DPW so we retrained D3DP jointly on Human 3.6M and MPII and tested on 3DPW.

| Baseline | CHAMP-Naive | CHAMP-Naive-Agg | CHAMP | CHAMP-Agg | D3DP | CameraPose | DiffPose |
|---|---|---|---|---|---|---|---|
| **PA-MPJPE (mm)** | 64.2 | 63.7 | 61.5 | 60.8 | 64.3 | 63.3 | 64.0 |

Table 6: Baseline Methods Trained on Human3.6M and MPII and Tested on 3DPW with PA-MPJPE Metric

As we can see, CHAMP outperforms other baselines by a noticeable margin. Even though 3DPW was not seen during training, our method achieves competitive performance. Moreover, we see a similar trend that J-Agg's edge is less than CP refinement: from CHAMP-Naive to CHAMP-Naive-Agg, the error is reduced by 0.5 mm, but it gets reduced by 2.7mm going from CHAMP-Naive to CHAMP, where CP refinement is added. This again indicates the importance of using CP to aggregate the hypotheses.

## M    MODULAR DESIGN

CHAMP uses the idea of attention mechanism to encode both spatial and temporal information. This is inherited from the usage of the MixSTE - D3DP backbone, which uses a Transformer-based encoder. While prior work such as Zheng et al. (2021); Zhao et al. (2023) do not consider multi-hypothesis settings or uncertainty quantification, switching the transformer in D3DP backbone in CHAMP with Zheng et al. (2021) or Zhao et al. (2023) will result in a similar multi-hypothesis model that is able to quantify uncertainty via end-to-end CP.

Similarly, Ci et al. (2023) uses an innovative approach to learning probabilistic human poses via gradient flow diffusion. The key difference lies in the formulation of the diffusion process to generate multiple hypotheses. We show that CHAMP's framework is modular enough as we are able to replace the diffusion backbone with GFPose's gradient-based diffusion. Mehraban et al. (2024) uses GCNFormer to improve the modeling of local dependencies inherent in human pose sequences, which is shown to outperform transformers. The key difference is again the choice of backbone. We replace the human pose backbone in CHAMP with GCNFormer and show the improvement gained using the conformity score.

Thus, for the aforementioned baselines, we finetune them with the end-to-end score function proposed in our work and compare how much the MPJPE gets reduced. We show the improvement brought by using the end-to-end conformal score function on the Human 3.6M dataset below in Table 7.

| Baseline | PoseFormer | PoseFormer V2 | GFPose | MotionAGFormer | CHAMP |
|---|---|---|---|---|---|
| **MPJPE Reduction (mm)** | 1.5 | 1.2 | 1.7 | 1.8 | 2.3 |

Table 7: Using CHAMP's End-to-End Score Function on Different Backbones

