# OpenReview forum: "CHAMP: Conformalized 3D Human Multi-Hypothesis Pose Estimators"
_ICLR.cc/2025/Conference — ICLR 2025 Poster_

### Official Review · Reviewer_83JW · 2024-10-16

**Soundness:** 2
**Presentation:** 2
**Contribution:** 2
**Rating:** 6
**Confidence:** 5

**Summary:**

This paper proposes using a diffusion model to generate multiple hypotheses for temporal 2D to 3D pose estimation and introduces Conformal Prediction (CP) to select more plausible hypotheses and aggregate them. The method achieves competitive results on two pose estimation benchmarks.

**Strengths:**

1. The paper introduces the novel idea of using Conformal Prediction to select more plausible results from multiple hypotheses, which is a fresh perspective.
2. The quantitative experiments demonstrate that the hypotheses selected and aggregated using this strategy outperform the results without filtering.

**Weaknesses:**

1. The paper's contribution is to be discussed. First, the ambiguity in temporal 2D-to-3D pose estimation is not as significant as in the single-frame, single-view case. Additionally, generating multiple 3D hypotheses using diffusion models is already a widely used technique (DiffuPose (Choi et al., 2023) DiffPose (Gong et al., 2023) DiffPose (Feng et al., 2023) D3DP (Shan et al., 2023)). If the only contribution is the use of CP, its novelty might be limited. If the authors could show that the CP strategy also performs well in single-RGB input cases or demonstrates its effectiveness in single-image mesh estimation, it would significantly enhance the paper's contribution.
2. The quantitative results are not very solid. The improvement on 3DHP is quite marginal, making it unclear whether the improvement is due to random seed variation. These results do not convincingly support the method's effectiveness.
3. The visual results (e.g., Figure 10) do not show cases where severe occlusion leads to incorrect 2D estimates or cases where 3D depth errors occur. It’s important to demonstrate that CP can filter out incorrect hypotheses in such challenging cases, where multi-hypothesis generation is truly necessary. This ties back to the first weakness, as the temporal nature of the data may already provide additional constraints, reducing ambiguity. Therefore, the authors should explore more varied tasks to prove CP's efficacy, such as monocular pose estimation.

**Questions:**

Regarding the method, it is unclear how the approach handles situations where the 2D estimates are incorrect or when the 2D pose is truncated beyond the image boundaries.

---

> ### Author Response · Authors · 2024-11-25
> **Thank You. Please see author rebuttal below. (1/3)**
>
> We really appreciate Reviewer 83JW for providing their insightful comments and questions. We address them below.
>
> > If the authors could show that the CP strategy also performs well in single-RGB input cases or demonstrates its effectiveness in single-image mesh estimation, it would significantly enhance the paper's contribution.
>
> The reviewer has raised a very valid point: When demonstrating the effectiveness of CHAMP, one problem is that the ambiguity in video 3D pose estimation where temporal information is abundant is not as significant as in the single-frame case. Thus, we test whether the proposed CP pipeline could still work for single-frame image-based 3D pose estimation. We emphasize that this requires minimal change to our method pipeline as we can adjust the number of frames ($N$) to 1 from 243. We train our method as well as the baseline methods with $N=1$ and test on the same dataset. We document the performance in **Table 5 of Appendix J**. It should be expected that with fewer frames available, all tested baselines should result in worse performance than with multiple frames. Interestingly, as we can see in Table 5, CHAMP-Agg is able to achieve a more significant edge over the previous SOTA (DiffPose) in single-frame setting vs in multi-frame (3.4mm vs 2.1mm). Moreover, CHAMP achieves more improvement over the backbone D3DP in the single-frame setting than in the multi-frame setting. **Results suggest that with less temporal information available (i.e. when the ambiguity is more severe), CHAMP achieves even more competitive performance compared to baselines.**
>
> > The improvement on 3DHP is quite marginal, making it unclear whether the improvement is due to random seed variation. These results do not convincingly support the method's effectiveness.
>
> While the improvement over current SOTA is not huge, we did add an important component on top of current SOTA methods, namely, the uncertainty quantification part. Moreover, while most improvement from prior work comes from fundamental changes to the architectures of the network backbones, **CHAMP achieves a noticeable performance boost over its backbone (D3DP) solely from the use of an end-to-end nonconformity score (uncertainty) function and the use of CP *after* training without any substantial changes to the backbone itself.** As one can see in Table 1, CHAMP did have a substantial improvement over D3DP, which is the human reconstruction backbone of CHAMP. Thus, as CHAMP outperforms its base architecture (D3DP) and improves even more after using more sophisticated aggregation methods, we argue that the uncertainty quantification component is beneficial. Another note here is that the D3DP performance we put in the tables are obtained after the J-Agg procedure, so it is expected to be slightly better than taking a simple mean as aggregation, which is what CHAMP-Naive did). CHAMP-Naive does not use CP at all during test time and thus serves as the "lower bound" of our method and should be comparable to D3DP with mean aggregation (which is indeed the case if one refers to Table 1 of D3DP P-Agg rows). As shown in Fig. 7 of the Appendix, our method did achieve reliable uncertainty quantification (empirical coverage), which is a trait that no other baseline method could achieve.

---

> ### Author Response · Authors · 2024-11-25
> **Thank You. Please see author rebuttal below. (2/3)**
>
> > The quantitative results are not very solid. The improvement on 3DHP is quite marginal, making it unclear whether the improvement is due to random seed variation. These results do not convincingly support the method's effectiveness.
>
> We also evaluate CHAMP’s performance on a much harder dataset, 3DPW, in **Appendix L**. To ensure fairness, we compare single-frame models CHAMP ($N=1$), D3DP ($N=1$), DiffPose ($N=1$), and CameraPose [1] trained jointly on Human 3.6M and MPII without using 3DPW during training and test on 3DPW using the PA-MPJPE. CHAMP outperforms other baselines by a noticeable margin. **Even though 3DPW was not seen during training, our method achieves competitive performance.**
>
> > It’s important to demonstrate that CP can filter out incorrect hypotheses in such challenging cases, where multi-hypothesis generation is truly necessary.
>
> We clarify that CHAMP assumes 2D keypoints are provided and improving the quality of 2D keypoints is not in the scope of our work. However, it would still be interesting to see how CHAMP behaves under ambiguous 2D keypoints (i.e., 2D keypoints with large uncertainty from truncation and occlusion). To address the reviewer’s comments, we evaluated CHAMP ($N=1$) performance in **harder settings in Appendix K**. The main failure mode/difficulty of CHAMP comes from wrong/uncertain 2D keypoints predictions. Thus, we categorize such large input uncertainty as follows:
> - Truncation in 2D RGB input.
> - Wrong 2D keypoints prediction.
> - Self-occlusion in 2D RGB input.
>
> While first two types large uncertainty, truncation in 2D RGB input (which gives inaccurate 2D keypoints) and wrong 2D keypoints, are not the focus of our work (since we assume 2D keypoints from some off-the-shelf method), it would be interesting to see how CHAMP behaves when such uncertainty occurs. To test this, and to make the task even more challenging, we use the single-frame model suggested by Reviewer 83JW ---this allows us to isolate the temporal effects and focus on uncertainty in 2D RGB inputs only, as the temporal nature of the data may already provide additional constraints, reducing ambiguity. **Interestingly, CHAMP performs well in these cases and learns to discern occluded/wrong 2D keypoints from the good ones. Some interesting behaviors are observed —CHAMP tends to output more diverse proposals for ambiguous keypoints** (see below).
>
> - In **Fig. 13**, when the input is truncated, a decent 2D keypoint detector (such as HRNet and CPN) is still able to detect the truncated joints. As long as the joints’ 2D keypoints are available, CHAMP can run as usual.
>
> - Next, when the 2D keypoints are wrong to begin with, CHAMP does make mistakes and this is indeed one key failure mode of CHAMP. However, we observe interesting behavior that CHAMP tends to correct “unnatural” 2D keypoints output. Specifically, in **Fig. 14**, we select two images where the 2D keypoint detector makes mistakes. In the first image, the left leg keypoints are completely off. Interestingly, the proposed 3D keypoints are a lot more diverse for the left leg than other joints. After the CP refinement step, the left leg joints are still more diverse, covering a wider region than other joints’ outputs do. In the second image, the two hands keypoints are off. We again see a much noisier proposal set for the two joints. After the refinement, the two hands' outputs are noticeably more diverse. One possible reason for this interesting phenomenon could be that the 2D keypoints input itself is seen as ``unnatural'' compared to the training data, forcing the network to output more diverse 3D poses.
>
> - Under self-occlusion, where the human subject itself occludes the joints, CHAMP’s behaviors are more interesting. Note that this is different from truncation in that truncation is caused by external occlusion rather than self-occlusion.  In **Fig. 15**, while the 2D detector is able to differentiate the two joints and output two different 2D keypoints, they are very close and the 3D locations are ambiguous. CHAMP outputs noisier proposals for the occluded arm joints (elbow and hand). Interestingly, after the CP refinement, the elbow joint pose outputs are almost ``bimodal'' and the hand pose outputs are still a lot noisier than other joints. For the second image, the dancer's right leg is occluded by the left leg. Similarly, CHAMP outputs noisier proposals for the right knee and right foot both before and after the CP refinement. One possible reason for this interesting phenomenon is that the network intrinsically learned to output different poses when the 2D keypoints are close to each other, suggesting possible self-occlusion.

---

> ### Author Response · Authors · 2024-11-25
> **Thank You. Please see author rebuttal below. (3/3)**
>
> > Therefore, the authors should explore more varied tasks to prove CP's efficacy, such as monocular pose estimation.
>
> We emphasize that our method is indeed monocular, as we only take as input 2D keypoints estimated from monocular videos or images.

---

> ### Comment · Reviewer_83JW · 2024-11-25
> **Thanks for the response**
>
> Thank you for providing additional results. I believe these results significantly enhance the quality of the paper, and I have therefore raised my score to 5. However, my primary concern remains the **novelty and contribution** of the work, particularly given that the current approach seems to involve directly applying an existing CP method to a 3D Human Pose Estimation (3D HPE) task.
>
> The authors have overlooked a considerable number of state-of-the-art studies in 3D HPE, particularly in the realm of probabilistic methods, which have been extensively explored in recent years. Since the primary contribution lies in its application to 3D HPE, the authors must provide a more comprehensive discussion and comparison with prior works in this area to better situate their contribution within the existing body of literature.
>
> The following works should be thoroughly discussed and compared against, including but not limited to:
> 1. 3D Human Pose Estimation with Spatial and Temporal Transformers (ICCV 2021)
> 2. GFPose: Learning 3D Human Pose Prior With Gradient Fields (CVPR 2023)
> 3. PoseFormerV2: Exploring Frequency Domain for Efficient and Robust 3D Human Pose Estimation (CVPR 2023)
> 4. MotionBERT: A Unified Perspective on Learning Human Motion Representations (ICCV 2023)
> 5. MotionAGFormer: Enhancing 3D Human Pose Estimation With a Transformer-GCNFormer Network (WACV 2024)
> 6. ScoreHypo: Probabilistic Human Mesh Estimation with Hypothesis Scoring (CVPR 2024)

---

> ### Comment · Reviewer_83JW · 2024-12-03
>
> Thank you for providing the additional discussion on related works. These clarifications help better articulate the contributions of CHAMP, and I appreciate the detailed analysis of its differences from prior approaches. I am raising my score to 6, as these discussions strengthen the positioning of the work. I also recommend incorporating these points into the revised version. The proposed experiments will further demonstrate CHAMP's potential as a modular framework that can adapt to different backbones and tasks, which would significantly enhance its impact.

---

### Official Review · Reviewer_aAY1 · 2024-11-03

**Soundness:** 3
**Presentation:** 2
**Contribution:** 2
**Rating:** 6
**Confidence:** 5

**Summary:**

This paper focuses on 3D pose estimation from 2D pose inputs in a multi-hypothesis sequence-to-sequence way. Following previous methods, they also adopt a diffusion model to regress the 3D pose hypotheses from sequential 2D pose inputs. Different from previous work, this paper proposes to learn a score model along with the regression model to predict the confidence of multiple regressed hypotheses. The scores of multiple hypotheses could be used to filter out the outliers to refine prediction set.
The eperiments are mostly conducted on two benchmarks, Human3.6M and MPI-INF-3DHP. Compared with directly averaging all multi-hypothesis predictions, averaging filtered predictions with the proposed score model reduces 1.5mm MPJPE on Human3.6M, while selecting the 3D pose whose 2D projection is more consistent with the 2D pose further reduces 1.5mm MPJPE on Human3.6M.

**Strengths:**

1. Learning the uncertainty of multi-hypothesis 3D human pose regression seems to be reasonable. Aside from building the distrubution of 3D human poses, how to select the best ones is an interesting research topic.
2. Designs in scoring the hypotheses via conformal prediction are subtle. Detailed ablation study further reveals the influence of different parameters.
3. The description of method is clear. Especially, Fig. 2 provides a clear overview of the relationship between different designs.

**Weaknesses:**

1. Experiments.
Tab. 1 and Tab. 2 show the improvement, compared with CHAMP-Naive, via introducing comformal prediction (CHAMP) and further selecting the 3D pose via measuring the 2D projection and 2D pose (CHAMP-Agg).  However, one important experiment seems to be missing here, CHAMP-Naive + -Agg. How well it performs if the proposed comformal prediction is not used?
This experiment will clearly show how important the proposed comformal prediction is.
Besides, the quantative eperiments are only conducted on indoor benchmarks. The qualitative results are relatively limited.  Performance on in-the-wild benchmarks, such as 3DPW, is not clear.

2. The performance improvement is not significant compared with the baseline SOTAs, which might effect the value of this work.

**Questions:**

See the experiments mentioned above.

---

> ### Author Response · Authors · 2024-11-25
> **Thank You. Please see author rebuttal below. (1/2)**
>
> We really appreciate Reviewer aAY1 for providing their insightful comments and suggestions. We address them below.
>
> > However, one important experiment seems to be missing here, CHAMP-Naive + -Agg. How well it performs if the proposed comformal prediction is not used? This experiment will clearly show how important the proposed comformal prediction is.
>
>
> Adding **CHAMP-Naive-Agg** is indeed very insightful and we now have added and highlighted the new experiments in **Table 1 and 2**. Specifically, the CHAMP-Naive-Agg baseline makes sure the output is aggregated with the J-Agg method from D3DP instead of taking a simple average. To compare the effectiveness of using conformal predictions, we compare the improvement from using CHAMP-Naive to CHAMP-Naive-Agg vs. from CHAMP-Naive to CHAMP. Going from CHAMP-Naive to CHAMP-Naive-Agg, the MPJPE gets reduced by 0.5mm on Human 3.6M and 0.3mm on MPII 3DHP. From CHAMP-Naive to CHAMP, the MPJPE is reduced by 2.3mm on Human 3.6M and 0.8mm on MPII 3DHP. **Thus, adding CP refinement indeed brings more improvements, indicating the importance of conformal prediction here**.
>
> > The qualitative results are relatively limited.
>
> We show more qualitative results in the new contents in the Appendix, which exhibit really interesting behaviors. Specifically, it would be interesting to see how CHAMP behaves under ambiguous 2D keypoints (i.e., 2D keypoints with large uncertainty from truncation and occlusion).  To address the reviewer’s comments, we evaluated CHAMP’s performance qualitatively in **harder settings** in Appendix K. The main failure mode/difficulty of CHAMP comes from wrong/uncertain 2D keypoints predictions. Thus, we categorize such large input uncertainty as follows:
> - Truncation in 2D RGB input.
> - Wrong 2D keypoints prediction.
> - Self-occlusion in 2D RGB input.
>
> While the first two types of large uncertainty, truncation in 2D RGB input (which gives inaccurate 2D keypoints) and wrong 2D keypoints, are not the focus of our work (since we assume 2D keypoints from some off-the-shelf method), it would be interesting to see how CHAMP behaves when such uncertainty occurs.
>
> To test this, and to make the task even more challenging, we use the single-frame model suggested by Reviewer 83JW ---this allows us to isolate the temporal effects and focus on uncertainty in 2D RGB inputs only, as the temporal nature of the data may already provide additional constraints, reducing ambiguity. **Interestingly, CHAMP performs well in these cases and learns to discern occluded/wrong 2D keypoints from the good ones. Some interesting behaviors are observed —CHAMP tends to output more diverse proposals for ambiguous keypoints** (see below).
>
> - In **Fig. 13**, when the input is truncated, a decent 2D keypoint detector (such as HRNet and CPN) is still able to detect the truncated joints. As long as the joints’ 2D keypoints are available, CHAMP can run as usual.
>
> - Next, when the 2D keypoints are wrong to begin with, CHAMP does make mistakes and this is indeed one key failure mode of CHAMP. However, we observe an interesting behavior where CHAMP tends to correct “unnatural” 2D keypoints output. Specifically, in **Fig. 14**, we select two images where the 2D keypoint detector makes mistakes. In the first image, the left leg keypoints are completely off. Interestingly, the proposed 3D keypoints are a lot more diverse for the left leg than other joints. After the CP refinement step, the left leg joints are still more diverse, covering a wider region than other joints outputs do. In the second image, the two hands keypoints are off. We again see a much noisier proposal set for the two joints. After the refinement, the two hands outputs are noticeably more diverse. One possible reason for this interesting phenomenon could be that the 2D keypoints input itself is seen as ``unnatural''
> compared to the training data, forcing the network to output more diverse 3D poses.
>
> - Under self-occlusion, where the human subject itself occludes the joints, CHAMP’s behaviors are more interesting. Note that this is different from truncation in that truncation is caused by external occlusion rather than self-occlusion.  In **Fig. 15**, while the 2D detector is able to differentiate the two joints and output two different 2D keypoints, they are very close and the 3D locations are ambiguous. CHAMP outputs noisier proposals for the occluded arm joints (elbow and hand). Interestingly, after the CP refinement, the elbow joint pose outputs are almost ``bimodal'' and the hand pose outputs are still a lot noisier than other joints. For the second image, the dancer's right leg is occluded by the left leg. Similarly, CHAMP outputs noisier proposals for the right knee and right foot both before and after the CP refinement. One possible reason for this interesting phenomenon is that the network intrinsically learned to output different poses when the 2D keypoints are close to each other, suggesting possible self-occlusion.

---

> ### Author Response · Authors · 2024-11-25
> **Thank You. Please see author rebuttal below. (2/2)**
>
> > Performance on in-the-wild benchmarks, such as 3DPW, is not clear.
>
> To address the reviewer’s comment, we evaluate CHAMP’s performance on 3DPW in **Appendix L**. We compare CHAMP with other baselines. To ensure fairness, we compare single-frame models CHAMP ($N=1$), D3DP ($N=1$), DiffPose ($N=1$), and CameraPose [1] trained jointly on Human 3.6M and MPII without using 3DPW during training and test on 3DPW using the PA-MPJPE. CHAMP outperforms other baselines by a noticeable margin. **Even though 3DPW was not seen during training, our method achieves competitive performance.**
>
> > The performance improvement is not significant compared with the baseline SOTAs, which might effect the value of this work.
>
> While the improvement over current SOTA is not huge, we did add an important component on top of current SOTA methods, namely, the uncertainty quantification part ---by checking the set membership using the learned uncertainty score, one could decide if the results can be trusted with probabilistic guarantees. Moreover, while most improvement from prior work comes from fundamental changes to the architectures of the network backbones, **CHAMP achieves a noticeable performance boost over its backbone (D3DP) solely from the use of an end-to-end nonconformity score (uncertainty) function and the use of CP after training without any substantial changes to the backbone itself**. As one can see in Table 1, CHAMP did have a substantial improvement over D3DP, which is the human reconstruction backbone of CHAMP. Thus, as CHAMP outperforms its base architecture (D3DP) and improves even more after using more sophisticated aggregation methods, we argue that the uncertainty quantification component is beneficial. Another note here is that the D3DP performance we put in the tables are obtained after the J-Agg procedure, so it is expected to be slightly better than taking a simple mean as aggregation, which is what CHAMP-Naive did). CHAMP-Naive does not use CP at all during test time and thus serves as the "lower bound" of our method and should be comparable to D3DP with mean aggregation (which is indeed the case if one refers to Table 1 of D3DP P-Agg rows). As shown in Fig. 7 of the Appendix, our method did achieve reliable **uncertainty quantification (empirical coverage)**, which is a trait that no other baseline method could achieve.
>
> [1] Yang, Cheng-Yen, et al. "Camerapose: Weakly-supervised monocular 3d human pose estimation by leveraging in-the-wild 2d annotations." Proceedings of the IEEE/CVF Winter Conference on Applications of Computer Vision. 2023.

---

> ### Comment · Reviewer_aAY1 · 2024-11-26
> **Thanks for making the experiments clear.**
>
> Via adding the results of CHAMP-Naive-Agg, the effects of introducing conformal predictions become more obvious. Besides, results on 3DPW furthur reveal how well the model perform on an in-the-wild benchmark. With all these efforts, the influence / improvement brought by the proposed conformal prediction has been well validated. Therefore, I upgrade the rate.

---

> ### Author Response · Authors · 2024-11-26
> **Thank you for your feedback and raising the score.**
>
> We sincerely appreciate your invaluable feedback on our work and for raising the score. We believe the 3D HPE task with CP holds significant potential, and still has considerable room for further development. We will continue to explore this direction in our future work.

---

### Official Review · Reviewer_rtwB · 2024-11-03

**Soundness:** 3
**Presentation:** 4
**Contribution:** 3
**Rating:** 8
**Confidence:** 4

**Summary:**

This work tackles the task of 2D-to-3D lifting of temporal human pose sequences through a diffusion-based multi-hypothesis approach. The novel aspect is applying the tools of conformal prediction to this task. That is, the authors propose to learn a conformity score function in an adversarial setup and simulate conformal prediction during training, directly minimizing the conformal set size (the inefficiency). Quantitative evaluation is performed on the well-known Human3.6M and MPI-INF-3DHP datasets, with additional qualitative evaluation on in-the-wild videos. Results show that filtering according to conformity scores and aggregating the results improves the metrics and achieves state-of-the-art results. Ablations also show that the end-to-end training of the scoring function is beneficial.

**Strengths:**

The application of conformal prediction to 3D human pose estimation is an interesting and original aspect. Multi-hypothesis 3D pose estimation has been a relevant research area due to ambiguities in the task, but practical methods are often not formulated in a proper probabilistic manner. Using conformal prediction techniques is a promising direction and taking a first step in this direction is relevant to the community.
The overview of the related works is extensive, and the comparison of them helps the reader to place this work in relation to them. The text is well-structured and easy to follow.
The authors use already existing techniques in good ways, for example using the already proven MixSTE model for denoising, incorporating adversarial training and conformal calibration.
Results on Human3.6M and MPI-INF-3DHP both demonstrate the benefit of the method.
Ablations and hyperparameter studies confirm the design choices.

**Weaknesses:**

The method is only evaluated on studio datasets, but not on outdoor ones, such as 3DPW and EMDB. Evaluation on such less restricted videos would be more convincing.
It is unclear whether CHAMP's learned conformity scoring performs better than the prior established 2D-joint projection based version (see question below).

**Questions:**

I don't fully understand the CHAMP-Agg method. Is this using J-Agg on the full set of hypotheses, or does it also use the learned scoring function in some way? How should we interpret the fact that "CHAMP-Agg" is better than "CHAMP"? Does it mean that the proposed learned conformity scoring does not perform better than the prior method of aggregation proposed by Shan et al. (2023)?

To the paragraph after Eq. 5: How can we interpret the "true value to be included in" the conformal set in case of continuous values like 3D coordinates? In case of classification this makes more sense, but perhaps the language has to adjusted here.
What 2D pose estimator is used in the main experiments, or is the ground truth used?
Why are values missing from Table 3?

Smaller issues:
GAN is cited as the CACM 2020 paper, but perhaps the original NIPS 2014 version is better suited for citation.
In Eq 9, perhaps Temperature could be denoted by some other symbol because mathcal{T} and tau look very similar visually (except for size). Similarly I would not recommend relying on the subtle difference between C and mathcal{C} for important differences (as noted in the footnote of page 6), a hat, tilde or similar feature may be more visible.

Page 7: Human3.6M is no longer the "largest indoor dataset for 3D human pose estimation". There are many larger ones such as DNA-Rendering.
It should be stated explicitly that E2E refers to "end-to-end".

---

> ### Author Response · Authors · 2024-11-24
> **Thank You. Please see author rebuttal below.**
>
> We sincerely appreciate Reviewer rtwB’s comments and suggestions. We address their comments below.
>
> > The method is only evaluated on studio datasets, but not on outdoor ones, such as 3DPW and EMDB. Evaluation on such less restricted videos would be more convincing
>
> To address the reviewer’s comments, we evaluated CHAMP’s performance on 3DPW in **Appendix L**. We compare CHAMP ($N=1$), D3DP ($N=1$), DiffPose ($N=1$), and CameraPose [1] trained jointly on Human 3.6M and MPII without using 3DPW during training and test on 3DPW using the PA-MPJPE. CHAMP outperforms other baselines by a noticeable margin. Even though 3DPW was not seen during training, our method achieves competitive performance.
>
> > I don't fully understand the CHAMP-Agg method. Is this using J-Agg on the full set of hypotheses, or does it also use the learned scoring function in some way? How should we interpret the fact that "CHAMP-Agg" is better than "CHAMP"? Does it mean that the proposed learned conformity scoring does not perform better than the prior method of aggregation proposed by Shan et al. (2023)?
>
> We apologize for the confusion about the approach used in CHAMP-Agg. CHAMP-Agg is using J-Agg aggregation method on the refined hypotheses set after running CP. Instead of using weighted averages to aggregate all refined proposals, we use J-Agg to aggregate them based on joint locations. It is expected that CHAMP-Agg is better than CHAMP because —**on the same refined hypotheses set**— CHAMP simply takes an average while CHAMP-Agg uses more sophisticated joint-level aggregation.
>
> > To the paragraph after Eq. 5: How can we interpret the "true value to be included in" the conformal set in case of continuous values like 3D coordinates? In case of classification this makes more sense, but perhaps the language has to adjusted here.
>
> We have changed the wording after Eq. 5 and for regression-based tasks, we say “set membership”. This is a subtle wording difference in that the set is defined via checking the conformity score with respect to the threshold, instead of explicitly constructing a set from discrete values. Moreover, the set here, **whose membership is easily checked using the conformity score**, can be constructed, and continuous values can also be included by calculating their conformity scores to check the set membership.
>
> > What 2D pose estimator is used in the main experiments, or is the ground truth used?
>
> In all of our main quantitative experiments we follow previous work and use 2D keypoints detected by CPN [2].
>
> > Why are values missing from Table 3?
>
> We apologize for the confusion in Table 3. Some authors did not report those metrics and we did not have the time to rerun the experiments. We now have run the experiments after submission and have filled in the numbers.
>
> > Smaller issues...
>
> We appreciate the reviewer for pointing out the notation and citation errors. We have fixed them and highlighted the changes.
> Similarly, we have changed the wording for Human 3.6M description, clarified E2E definition, and highlighted the changes.
>
> [1] Yang, Cheng-Yen, et al. "Camerapose: Weakly-supervised monocular 3d human pose estimation by leveraging in-the-wild 2d annotations." Proceedings of the IEEE/CVF Winter Conference on Applications of Computer Vision. 2023.
>
> [2] Yilun Chen, Zhicheng Wang, Yuxiang Peng, Zhiqiang Zhang,
> Gang Yu, and Jian Sun. Cascaded pyramid network for multiperson pose estimation. In Proceedings of the IEEE Conference on Computer Vision and Pattern Recognition (CVPR),
> pages 7103–7112, 2018.

---

> > ### Comment · Reviewer_rtwB · 2024-11-26
> >
> > I appreciate the response. The additional experiment on 3DPW helps confirm the generalization capability, and the other answers helped clarify my questions. I think introducing the conformal prediction framework to this task and showing that it helps to include it in the training phase is a significant enough contribution and has many potential extension directions. I think it is not a problem to first tackle the simpler pose lifting task and then take on others such as image-based mesh recovery etc. I therefore maintain my score and think that it is a work that is worth the community's attention.

---

> > > ### Author Response · Authors · 2024-11-26
> > > **Thank you for acknowledging the significance of CHAMP!**
> > >
> > > We sincerely thank you for your valuable suggestions and feedback. We greatly appreciate your acknowledgment of our contributions and fully agree that **"it is a work that is worth the community's attention" and "has many potential extension directions."** We believe that 3D HPE with CP holds significant potential, and still has considerable room for further development. We will continue to explore this direction in our future work.

---

### Official Review · Reviewer_dk3Y · 2024-11-05

**Soundness:** 3
**Presentation:** 3
**Contribution:** 3
**Rating:** 6
**Confidence:** 4

**Summary:**

CHAMP presents a conformalized multi-hypothesis pose estimation pipeline. Based on diffusion model, the multiple pose hypothesis are refined by a differentiable comformal prediction module with an inefficiency loss. Experiments show that CHAMP achieves state-of-the-art results on several benchmarks.

**Strengths:**

1. The proposed conformal prediction is simple yet effective.
2. This paper is well-written and easy to understand.

**Weaknesses:**

1. CP removes the predictions with low confidence and visualisations also show after CP the predictions become more certain and accurate. However, the results are mainly shown with easy cases where occlusions barely present. I am wondering how CP performs on hard cases where occlusions, truncations exist?
2. It will be better if this method can be extended to image-to-3d pipelines.

**Questions:**

This paper is well-presented and validated. The only thing I concern is its performance on occluded or truncations where diverse multiple hypothesis are preferred.

---

> ### Author Response · Authors · 2024-11-24
> **Thank You. Please see author rebuttal below.**
>
> We sincerely appreciate Reviewer dk3Y’s comments and suggestions. We address their questions below.
> > However, the results are mainly shown with easy cases where occlusions barely present. I am wondering how CP performs on hard cases where occlusions, truncations exist?
>
> We clarify that CHAMP assumes 2D keypoints are provided and improving the quality of 2D keypoints is not in the scope of our work. However, it would still be interesting to see how CHAMP behaves under ambiguous 2D keypoints (i.e., 2D keypoints with large uncertainty from truncation and occlusion).  To address the reviewer’s comments, we evaluated CHAMP’s performance in **harder settings** in Appendix K. The main failure mode/difficulty of CHAMP comes from wrong/uncertain 2D keypoints predictions. Thus, we categorize such large input uncertainty as follows:
> - Truncation in 2D RGB input.
> - Wrong 2D keypoints prediction.
> - Self-occlusion in 2D RGB input.
>
> While the first two types of large uncertainty, truncation in 2D RGB input (which gives inaccurate 2D keypoints) and wrong 2D keypoints, are not the focus of our work (since we assume 2D keypoints from some off-the-shelf method), it would be interesting to see how CHAMP behaves when such uncertainty occurs.
>
> To test this, and to make the task even more challenging, we use the single-frame model suggested by Reviewer 83JW ---this allows us to isolate the temporal effects and focus on uncertainty in 2D RGB inputs only, as the temporal nature of the data may already provide additional constraints, reducing ambiguity. **Interestingly, CHAMP performs well in these cases and learns to discern occluded/wrong 2D keypoints from the good ones. Some interesting behaviors are observed —CHAMP tends to output more diverse proposals for ambiguous keypoints** (see below).
>
> - In **Fig. 13**, when the input is truncated, a decent 2D keypoint detector (such as HRNet and CPN) is still able to detect the truncated joints. As long as the joints’ 2D keypoints are available, CHAMP can run as usual.
>
> - Next, when the 2D keypoints are wrong to begin with, CHAMP does make mistakes and this is indeed one key failure mode of CHAMP. However, we observe an interesting behavior where CHAMP tends to correct “unnatural” 2D keypoints output. Specifically, in **Fig. 14**, we select two images where the 2D keypoint detector makes mistakes. In the first image, the left leg keypoints are completely off. Interestingly, the proposed 3D keypoints are a lot more diverse for the left leg than other joints. After the CP refinement step, the left leg joints are still more diverse, covering a wider region than other joints outputs do. In the second image, the two hands keypoints are off. We again see a much noisier proposal set for the two joints. After the refinement, the two hands outputs are noticeably more diverse. One possible reason for this interesting phenomenon could be that the 2D keypoints input itself is seen as ``unnatural''
> compared to the training data, forcing the network to output more diverse 3D poses.
>
> - Under self-occlusion, where the human subject itself occludes the joints, CHAMP’s behaviors are more interesting. Note that this is different from truncation in that truncation is caused by external occlusion rather than self-occlusion.  In **Fig. 15**, while the 2D detector is able to differentiate the two joints and output two different 2D keypoints, they are very close and the 3D locations are ambiguous. CHAMP outputs noisier proposals for the occluded arm joints (elbow and hand). Interestingly, after the CP refinement, the elbow joint pose outputs are almost ``bimodal'' and the hand pose outputs are still a lot noisier than other joints. For the second image, the dancer's right leg is occluded by the left leg. Similarly, CHAMP outputs noisier proposals for the right knee and right foot both before and after the CP refinement. One possible reason for this interesting phenomenon is that the network intrinsically learned to output different poses when the 2D keypoints are close to each other, suggesting possible self-occlusion.
>
> > It will be better if this method can be extended to image-to-3d pipelines.
>
> We appreciate the reviewer for pointing out an interesting future direction. We are indeed working on the next version of CHAMP that works on full human mesh reconstruction based on SMPL learning. The same idea of using an end-to-end uncertainty score can be extended to both shapes and poses. Another direction we are working on is conformalized 3D object pose estimation. Again, the same idea can be applied.

---

> ### Author Response · Authors · 2024-12-02
>
> Given that the discussion phase is quickly passing, we would like to know if our response has addressed your concerns. If you have any further questions or suggestions, we would be more than happy to continue the discussion. Thank you again for your constructive feedback, and we look forward to hearing from you.

---

> > ### Comment · Reviewer_dk3Y · 2024-12-03
> >
> > Sorry, I made a mistake that my comment is not visible to authors. Here are the original comments.
> >
> > Thank the authors for your explanation and additional experimental results! I really appreciate that Appendix K gives a thorough demonstration of CHAMP's performance under different situations. I have one follow-up question: is the 2D input composed of uncertainty and 2D poses? if no, how CHAMP can recognize different scenarios and give more diverse 3D poses when joints are occluded?
> >
> > I also share the same concern with Reviewer 83JW. Since the CP is not limited to the lifting pipeline, it will be less novel to only show CP's improvement over the lifting pipeline, which also constraints the ability to predict invisible 3D poses due to the lack of image evidence. So, currently I tend to remain my score and I am looking forward to the authors' reply!

---

> ### Author Response · Authors · 2024-12-03
>
> We appreciate Reviewer dk3Y for the questions. Regarding the 2D input, the occlusion happens when two joints are close to each other in pixel space. "Uncertainty in 2D", however, is more subtle. As explained in the rebuttal text, our conjecture is that the wrong/occluded/uncertain 2D inputs are "unnatural" compared to the training data (i.e. not conforming to the training set), encouraging the model to output diverse samples. This is an interesting behavior exhibited in in-the-wild samples.
>
> We appreciate the reviewer for pointing out an intriguing future direction. While CHAMP assumes that 2D keypoints are provided (as per previous works), we argue that uncertainty-augmented prediction from images can be straightforwardly achieved using the proposed learned conformity score method. The idea of using end-to-end uncertainty (conformity score) learned with pose estimation can be easily extended to image-to-human-mesh (SMPL learning) pipelines. We are currently implementing the next version of CHAMP which works on predicting human shape and pose directly from RGB images. Preliminary results suggest a similar story, that the learned conformity score can also improve the performance of SMPL prediction directly from images. **The learned CP is almost modular and can be plugged into human mesh reconstruction from image pipelines** and further improve the SMPL learning performance without substantial changes to the mesh prediction backbone.
>
> We hope our explanation clears up the concerns and questions. Please let us know if you have further questions.

---

### Author Response · Authors · 2024-11-24
**Meta Rebuttal - Addressing Common Questions and Brief Summary of Changes**

Once again, we sincerely appreciate the reviewers’ efforts to review our work. We have addressed the comments and suggestions and ran new experiments to address the points raised by the reviewers. We have **updated the manuscript PDF and highlighted the new changes** to the content. Some common questions raised by the reviewers were summarized in this section, and individual reviewers’ questions were addressed in the comments below.

### **Common Questions**:
- **CHAMP performance under “difficult” cases such as truncation and occlusion (raised by Reviewer dk3Y and Reviewer 83JW) in 2D inputs**: We clarify that CHAMP assumes 2D keypoints are provided and improving the quality of 2D keypoints is not in the scope of our work. However, it would still be interesting to see how CHAMP behaves under ambiguous 2D keypoints (i.e., 2D keypoints with large uncertainty from truncation and occlusion).  We selected both synthetically-truncated RGB inputs and self-occluded RGB inputs. To make the task more difficult, as suggested by Reviewer 83JW, we trained another model using $N=1$ (frame number = 1) to eliminate the constraints/disambiguation from the temporal effects. Interestingly, CHAMP performs well in these cases and learns to discern occluded/wrong 2D keypoints from the good ones. Some interesting behaviors are observed —CHAMP tends to output more diverse proposals for ambiguous keypoints. (Appendix K)
- **CHAMP’s performance on harder datasets such as 3DPW (raised by Reviewer rtwB and aAY1)**: We were able to test our method on 3DPW after the submission period. Following previous work, we test on 3DPW as an out-of-distribution generalization task, by training jointly on Human 3.6M and MPII. CHAMP performs well in this task and beats baseline methods. (Appendix L).
- **CHAMP’s performance compared to current SOTAs (raised by Reviewer aAY1 and Reviewer 83JW)**: While the performance increase over SOTAs is not huge, the improvement is quite noticeable going from D3DP (CHAMP’s human reconstruction backbone) to CHAMP. Our main argument in this work is that Conformal Prediction with an end-to-end trained conformity score (uncertainty) is able to improve performance without substantial changes to the pose prediction backbone. We further provided **a new baseline, CHAMP-Naive-Agg**, indicating that the CP-based aggregation improvement is a lot more significant than using joint-based aggregation.
- **CHAMP's novelty (raised by Reviewer rtwB and 83JW)**: We maintain that the end-to-end conformity score in 3D HPE is a key novel contribution in that it is able to improve the 3D HPE performance without substantial backbone design changes. We added an important component on top of current SOTA methods, namely, the uncertainty quantification part ---**by checking the set membership using the learned uncertainty score, one could decide if the results can be trusted with probabilistic guarantees**. Moreover, the added CP component based on ConTr was initially designed for classification. To our knowledge, we were the first work that modified and applied it for a regression setting. Moreover, our method did achieve **reliable uncertainty quantification (empirical coverage), which is a trait that no other baseline method could achieve**.

### **Main Changes to the Manuscript**:
- **Section 4**: Notation and citation changes as suggested by Reviewer rtwB.
- **Section 5**: Added new CHAMP-Naive-Agg experiment suggested by Reviewer aAY1.
- **Section 5**: E2E clarification and missing baseline performance suggested by Reviewer rtwB.
- **Appendix J**: Single-frame variation of CHAMP suggested by Reviewer 83JW
- **Appendix K**: Performance when large uncertainty in 2D keypoints is present, suggested by Reviewer dk3Y and Reviewer 83JW.
- **Appendix L**: 3DPW performance requested by Reviewer rtwB and aAY1.

---

### Comment · Reviewer_83JW · 2025-03-05
**Concern Regarding the Omission of Previously Discussed Points in the Final Version**

It is concerning that the authors have not incorporated the key points we discussed during the discussion phase into the final version, despite their explicit commitment to doing so. Given the strong connections between CHAMP and prior works, our discussion emphasized the need for a more thorough comparison and clearer positioning. These clarifications were essential for distinguishing CHAMP’s contributions and ensuring that its novelty is properly articulated.

Since the authors had explicitly acknowledged the importance of these discussions and committed to incorporating them, it is disappointing that they are not reflected in the final version. Addressing these points would have significantly improved the clarity and impact of the paper.

---

> ### Public Comment · ~Harry_Zhang2 · 2025-03-05
>
> Hi reviewer, we did provide an extra section in Appendix M.

---

> > ### Comment · Reviewer_83JW · 2025-03-11
> >
> > At my request, the authors added a section in Appendix M. Yet, it doesn’t fulfill their rebuttal promise of a ''We are already looking into the works you mentioned and will definitely have a stronger discussion on related work in the final version.
> > While we are still implementing the suggested baselines and setting up the tests for fair comparisons, we provide a brief summary of the differences between CHAMP and the 6 mentioned related methods. We hope this will better situate the contributions of CHAMP. We will definitely add the numbers to the final version of the paper when we are done with implementations.''
> >
> > Trusting their commitment, I raised my score twice, and I’m disappointed by the shortfall. At minimum, the related work section in the main text needs revision to include key related works (e.g., GFPose, ScoreHypo).

---

> > > ### Public Comment · ~Harry_Zhang2 · 2025-03-11
> > >
> > > hi reviewer, we have added the requested work to the RW section. we also added new experiments for MotionAGFormer backbone.

---

### Meta-Review · Area_Chair_jsus · 2024-12-21

**Metareview:**

This paper addresses the task of 2D-to-3D lifting of temporal human pose sequences through a diffusion-based multi-hypothesis approach. It introduces Conformal Prediction (CP) to select more plausible hypotheses and aggregate them.  The method learns a conformity score function in an adversarial setup and simulates conformal prediction during training, directly minimizing the conformal set size (the inefficiency).  Applying conformal prediction to 3D human pose estimation is an interesting; in particular, in the context of selecting more plausible results from multiple hypotheses.  The proposed uncertainty quantification component will be beneficial to the community. One the other hand, the reviewers raised concerns regarding insufficient validation, unclear explanation of the proposed method, and lacking comprehensive discussion on the positioning and contribution of the presented work against prior works. To demonstrate the generalization capability of the method, evaluation on in-the-wild datasets (where occlusion, truncations exist) such as 3DPW and EMDB is required.  It is also suggested that the effectiveness of the CP strategy in single-RGB input cases should be demonstrated to enhance the contribution. The authors have provided additional experiments to address the concerns on experiments in the rebuttal with arguing in-depth analysis of the results. The authors’ rebuttal and following discussion between the authors and the reviewers have resolved the raised concerns, leading to clear contribution of the paper with convincing validation.  The reviewers are all positive for the paper.  This paper should be accepted, accordingly.

**Additional Comments On Reviewer Discussion:**

See above.

---

### Decision · Program_Chairs · 2025-01-22

Accept (Poster)